# Exploiting Completeness Perception with Diffusion Transformer for Unified 3D MRI Synthesis

## Abstract

Missing data problems, such as missing modalities in multi-modal brain MRI and missing slices in cardiac MRI, pose significant challenges in clinical practice. Existing methods rely on external guidance to supply detailed missing-state information for instructing generative models to synthesize missing MRIs. However, manual indicators are not always available or reliable in real-world scenarios due to the unpredictable nature of clinical environments. Moreover, these explicit masks are not informative enough to provide guidance for improving semantic consistency. In this work, we argue that generative models should infer and recognize missing states in a self-perceptive manner, enabling them to better capture subtle anatomical and pathological variations. Towards this goal, we propose CoPeDiT, a shared **completeness-perception** framework for 3D MRI synthesis, following a common conditioning strategy with task-specific instantiations for different missing-data scenarios. Specifically, we incorporate dedicated pretext tasks into our tokenizer, CoPeVAE, empowering it to learn completeness-aware discriminative prompt tokens, and design MDiT3D, a specialized diffusion transformer architecture for 3D MRI synthesis that effectively uses the completeness-aware prompt tokens as guidance to enhance semantic consistency in 3D space. Comprehensive evaluations on three large-scale MRI datasets demonstrate that CoPeDiT consistently improves upon state-of-the-art methods across diverse missing patterns, yielding high-fidelity and structurally consistent MRI synthesis. Our code is available at https://anonymous.4open.science/r/CoPeDiT-B0E9/.

## 1 Introduction

Magnetic resonance imaging (MRI) provides crucial anatomical and pathological insights, particularly through multi-modal brain and volumetric cardiac scans Dickinson et al. (2013); Lustig et al. (2007); Dayarathna et al. (2024); Manna et al. (2024). However, real-world clinical MRIs frequently suffer from missing data, including absent brain modalities and missing cardiac slices, due to limited scan times, image corruption, or protocol variations Wang et al. (2025b); Paproki et al. (2024); Zhao & Shen (2025).

To address this, generative models have been developed to infer missing data from observed inputs Ibrahim et al. (2025); Ferreira et al. (2024). Existing paradigms rely on auxiliary embeddings, e.g., binary mask codes, as prior knowledge to encode missing patterns (e.g., severity, type, and position) Liu et al. (2023); Hao et al. (2024); Kim & Park (2024); Cho et al. (2024); Meng et al. (2024). Nonetheless, these hand-crafted masks only indicate missing locations, without adequately characterizing the actual incomplete state of the input. This causes three practical limitations. First, because missing patterns vary across hospitals, scanners, and acquisition settings, enumerating them with predefined masks is unrealistic in real deployments Wang et al. (2023); Rui et al. (2025). Second, the resulting condition is insensitive to modality-specific and spatially varying context, making models less robust to unseen incomplete patterns and prone to degraded generalization Lee et al. (2023); Ke et al. (2025); Wenderoth et al. (2025); Hamamci et al. (2024); Azad et al. (2025); Pan et al. (2025). Third, because binary masks carry limited semantics, they provide rigid and insufficiently informative guidance, which can weaken spatial alignment and semantic consistency during synthesis Qiu et al. (2023); Hu et al. (2023); Shin et al. (2025); Wu et al. (2025).

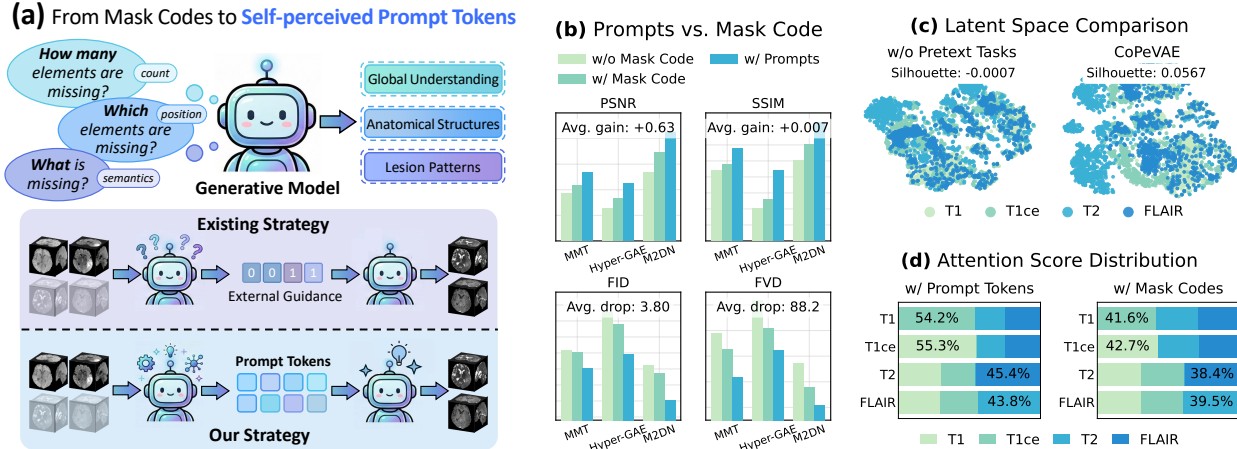

Figure 1: **(a)** We shift the paradigm from mask-dependent guidance to autonomous completeness perception via prompt token generation. This self-perceptive mechanism offers significant advantages: **(b)** quantitative plug-and-play performance gains across existing baselines; **(c)** substantially more discriminative latent representations compared to training without pretext tasks; and **(d)** enhanced semantic attention alignment between correlated modalities (e.g., T1-T1ce, T2-FLAIR) compared to traditional binary mask codes.

Intuitively, generative models should infer and detect the incomplete state spontaneously, rather than relying on externally provided guidance Hu et al. (2023); Graikos et al. (2024). Motivated by this, we pose a central question: *Can we empower the model with the ability to perceive missing states on its own?* In light of this, we exploit an underexplored property in medical imaging, i.e., **'*completeness perception*'**, to enhance flexibility and generalizability under diverse missing MRI conditions, as illustrated in Fig. 1a. Our fundamental insight is to enable the generative model to recognize the fine-grained incomplete state information in a self-perceptive manner, and to leverage this understanding as internal prompt tokens to guide the generation process. Our empirical results suggest that, for diffusion models, such self-guided prompt tokens can serve as an effective alternative to manually defined masks and provide more informative guidance signals (Fig. 1b). The main reason lies in the fact that this self-perceptive strategy encourages the model to learn global and local anatomical structures and lesion patterns at coarse and fine levels, enabling more semantically coherent generation of the missing MRI regions (Figs. 1c, 1d) Liang et al. (2022).

Driven by our motivation, we propose CoPeDiT, a 3D latent diffusion model (LDM) framework built upon a shared completeness-perception design paradigm for 3D MRI synthesis. Here, our "unified" denotes a shared missing-data synthesis framework with a common prompt-token-based conditioning and diffusion generation pipeline, while CoPeDiT is trained with task-specific adaptations for brain missing-modality and cardiac missing-slice synthesis. Technically, our framework builds on two core components: **(i)** Unlike prior approaches that require explicit missing indicators, a novel tokenizer with a completeness-perception function, CoPeVAE, is proposed to autonomously assess the integrity of modalities or volumes through tailored self-supervised pretext tasks. By detecting anatomical structures and variations in lesion patterns, CoPeVAE develops a comprehensive understanding of 3D MRIs. This enables CoPeDiT to reduce reliance on manual intervention while improving flexibility and adaptability, enhancing the method's autonomy and improving its feasibility with diverse missing patterns. **(ii)** A task-specific diffusion transformer instantiation, MDiT3D, is developed as a dependency-aligned conditioning interface for completeness-aware prompt tokens in 3D MRI synthesis. Rather than introducing a fundamentally new DiT paradigm, MDiT3D adapts 3D tokenization, axis-specific attention, dynamic reshaping, and targeted prompt token injection to the task-relevant dependencies of volumetric MRIs, including inter-modal relationships in brain MRI and through-plane continuity in cardiac MRI. This design provides the proper pathway for our learned prompt tokens to influence the generative process, allowing the prompt tokens to propagate observed modality- or slice-specific cues along anatomically meaningful dependencies during diffusion. As a result, MDiT3D enables

more reliable synthesis of missing modalities or slices while better preserving structural consistency in high-dimensional 3D data. Incorporating the above two innovations, our architecture not only enables adaptive self-guidance synthesis, but also demonstrates improved structural coherence and enhanced preservation of fine-grained anatomical details. Our main contributions are summarized as follows:

- We propose CoPeDiT, a unified framework with task-specific variants for 3D brain and cardiac missing MRI synthesis, sharing a common self-supervised semantic prompting paradigm and diffusion-based generation pipeline without requiring explicit external indicators as guidance.

- We equip our tokenizer, CoPeVAE, with carefully designed pretext tasks to learn completeness-aware prompt tokens that encode global missing severity, fine-grained missing positions, and modality/slice-specific semantic priors, providing more informative guidance than binary mask codes.

- We present MDiT3D, a tailored diffusion transformer for 3D MRI synthesis, designed as a dependency-aligned conditioning interface to effectively inject the learned semantic prompt tokens into task-relevant generation blocks.

- Extensive experiments on three datasets demonstrate that our model achieves consistently competitive performance compared with state-of-the-art (SOTA) methods.

## 2 Related Work

**Medical Image Generation.** Medical image generation has been widely studied for data augmentation Hamamci et al. (2024); Zhao et al. (2025), reconstruction Yu et al. (2025); Xiao et al. (2025); Liu et al. (2024), and image completion Rassmann et al. (2025); Liu et al. (2023); Song et al. (2026) in clinical imaging workflows. Earlier methods based on Generative Adversarial Networks (GANs) improved realism but frequently faced limitations in training stability and mode collapse, restricting their fidelity, especially when scaling to complex, high-dimensional 3D volumetric data Cao et al. (2023); Shao et al. (2025); Weng et al. (2024). To overcome these bottlenecks, diffusion-based models have emerged as a robust alternative. By offering stable training dynamics and superior mode coverage, they have achieved competitive performance in medical image synthesis and are increasingly favored in conditional settings guided by partial observations, anatomical priors, or auxiliary modalities Wang et al. (2025a); Zhao et al. (2025); Zhang et al. (2026); Guo et al. (2025); Yeganeh et al. (2025).

**MRI Synthesis.** Recent unified MRI synthesis methods predominantly utilize generative models, ranging from GANs Xia et al. (2021); Zhang et al. (2024; 2019a;b); Yang et al. (2023); Sharma & Hamarneh (2020) and Transformers Liu et al. (2023; 2021; 2025) to advanced diffusion models Qiu et al. (2025); Yeganeh et al. (2025); Meng et al. (2024); Song et al. (2026). While these approaches successfully capture complex inter-modal dependencies to impute missing data, they inherently rely on externally provided masks to explicitly encode missing patterns in randomly incomplete scenarios Liu et al. (2023); Meng et al. (2024); Wang et al. (2023); Azad et al. (2025). This manual guidance is often rigid and lacks informative semantic details about the actual incomplete state of the input. Furthermore, because missing patterns vary widely across different clinical environments, requiring predefined masks limits practical deployment. In contrast to these mask-dependent paradigms, CoPeDiT explores the self-perceptive capability of generative models to autonomously recognize data completeness. By learning to infer missing states internally, our approach reduces reliance on manual intervention, enabling more flexible and high-fidelity MRI synthesis.

**Diffusion Models.** Diffusion models, particularly LDMs Rombach et al. (2022), have demonstrated remarkable capabilities across vision tasks Ho et al. (2020); Lu et al. (2024); Ma et al. (2025); Yao et al. (2025). Recently, DiTs Peebles & Xie (2023) have emerged as powerful alternatives to traditional U-Net Ronneberger et al. (2015) backbones, achieving competitive performance in image synthesis. However, most existing medical image generation approaches still predominantly rely on U-Net architectures, leaving the potential of transformer-based LDMs largely underexplored in this domain Wang et al. (2025a); Nazir et al. (2025). To bridge this gap, we introduce MDiT3D, which replaces the U-Net backbone with a diffusion transformer. By incorporating task-specific architectural modifications, MDiT3D aligns diffusion modeling with 3D spatial context, inter-modal relationships, and through-plane continuity in volumetric MRI synthesis.

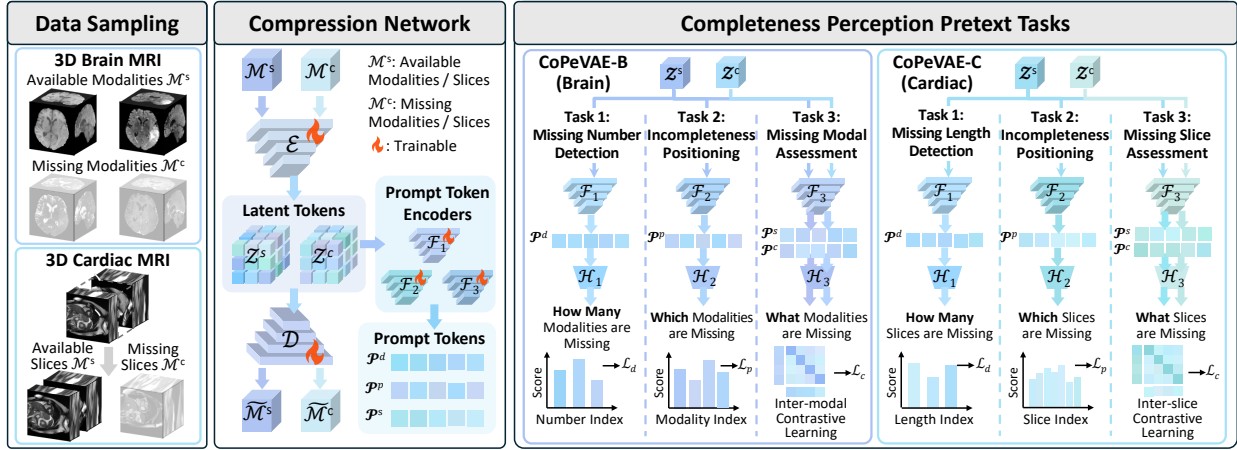

Figure 2: **The overview framework of CoPeVAE.** We implement two variants, CoPeVAE-B and CoPeVAE-C, with slight architectural modifications for brain and cardiac MRI synthesis tasks, respectively.

## 3 Methodology

### 3.1 Notations

To present brain missing-modality synthesis and cardiac missing-slice synthesis under a common notation, let $\mathcal{M} = \{x^i\}_{i=1}^m$ denote a complete 3D MRI sample comprising $m$ elements, which correspond to either modalities or slices depending on the task. We partition $\mathcal{M}$ into an available subset $\mathcal{M}^S = \{\mathbf{x}^{s_i}\}_{i=1}^s$ and a missing subset $\mathcal{M}^C = \{\mathbf{x}^{c_i}\}_{i=1}^c$, where $m = s + c$. The objective is to synthesize $\mathcal{M}^C$ from $\mathcal{M}^S$. Specifically, for **brain MRIs**, $\mathcal{M}^C$ contains randomly missing modalities; for **cardiac MRIs**, $\mathcal{M}^C$ consists of consecutive missing slices. To mimic real-world clinical environments, we evaluate randomly generated incomplete cases with varying missing counts and lengths.

### 3.2 Stage I: Completeness Perception Tokenizer

The core idea of CoPeVAE (Fig. 2) is that detecting the completeness of high-resolution MRI data enforces the model to perceive both global anatomy and local lesion patterns, thereby producing high-quality prompt tokens as diffusion guidance. Building upon VQGAN van den Oord et al. (2017); Esser et al. (2021), we deploy a 3D autoencoder jointly trained with self-supervised pretext tasks. Each task employs a prompt token encoder, denoted as $\mathcal{F}_1, \mathcal{F}_2$ and $\mathcal{F}_3$, to transform latent tokens (learned by the encoder $\mathcal{E}$) into low-dimensional prompt tokens. All prompt token encoders contain 3D Conv layers followed by spatial average pooling. Afterwards, task-specific projection heads are applied for multi-granular classification and contrastive learning.

To improve CoPeVAE's adaptability to diverse missing cases, we employ a dual-random sampling strategy, where both the number/length of missing elements and the modality types/slice positions are randomly sampled. Given a complete set $\mathcal{M}$, we randomly sample a missing count $c \in \{1, \ldots, m-1\}$, then uniformly select $c$ elements to form the missing subset $\mathcal{M}^C$, with the remainder forming the incomplete subset $\mathcal{M}^S$.

**Task 1. Missing Number/Length Detection.** Equipped with global contextual awareness, the tokenizer is capable of determining how many modalities/slices are missing in the incomplete input. This task aims to enable the tokenizer to identify the severity of incompleteness by perceiving the global context of MRIs, thereby learning modality/spatial attributes in a coarse-grained manner. We formulate this task as an $(m-1)$-class classification task and define the loss using cross-entropy loss as follows:

$$\mathcal{L}_d = \mathcal{L}_{\text{cls}}(\mathcal{H}_1(\mathcal{F}_1(\mathbf{z}^s))_c). \tag{1}$$

where $\mathcal{F}_1$ and $\mathcal{H}_1$ represent the prompt token encoder and the projection head, respectively. The learned prompt tokens $\mathbf{p}^d = \mathcal{F}_1(\mathbf{z}^s)$ are rich in information about the severity of the missing state.

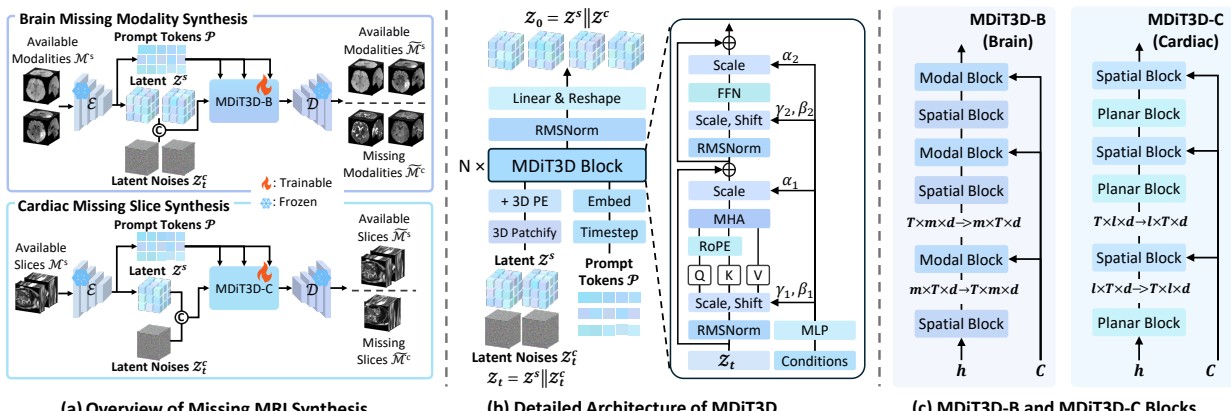

Figure 3: **Architecture of the MDiT3D framework.** We implement two variants with alternating blocks (Spatial/Modal for MDiT3D-B and Planar/Spatial for MDiT3D-C), using dynamic feature reshaping to model respective anatomical dependencies. Learned prompt tokens are injected via adaLN as conditional guidance. PE: Positional Embeddings; RoPE: Rotary Position Embeddings Su et al. (2024); Yang et al. (2025).

**Task 2. Incompleteness Positioning.** By identifying which modalities or slices are missing, CoPeVAE yields prompt tokens $\mathbf{p}^p = \mathcal{F}_2(\mathbf{z}^s)$ that capture semantically meaningful local properties. The motivation of this task is to drive the model to develop a finer-grained contextual understanding of subtle anatomical structures and detailed pattern variations. Although the missing position implicitly encodes the count, Task 1 learns a modality/slice-agnostic global magnitude prior that calibrates the conditioning strength, while this task provides discrete, spatially localized cues about the exact missing identity. The incorporation of the two tasks improves robustness to errors from either signal. This task is formulated as an $m$-class classification problem and is also optimized using the cross-entropy loss, defined as follows:

$$\mathcal{L}_p = \mathcal{L}_{\text{cls}}(\mathcal{H}_2(\mathcal{F}_2(\mathbf{z}^s))_I). \tag{2}$$

where $I$ denotes the index of missing type/position.

**Task 3. Missing Modality/Slice Assessment.** Motivated by the observation that modalities or slices from the same scan share more similar anatomical and textural context than those from different scans, we adopt an inter-modal/inter-slice contrastive learning scheme Radford et al. (2021) to serve as a missing data estimator. We take the incomplete tokens $\mathbf{z}^s$ as the anchor, the corresponding missing latent $\mathbf{z}^c$ from the same subject as positives, and tokens $\mathbf{z}^c_-$ from different subjects as negatives, yielding:

$$\mathcal{L}_c = -\log \frac{\varphi\left(\mathcal{H}_3(\mathbf{p}^s), \mathcal{H}_3(\mathbf{p}^c)\right)}{\varphi\left(\mathcal{H}_3(\mathbf{p}^s), \mathcal{H}_3(\mathbf{p}^c)\right) + \sum \varphi\left(\mathcal{H}_3(\mathbf{p}^s), \mathcal{H}_3(\mathbf{p}^c_-)\right)}. \tag{3}$$

where $\mathbf{p}^s = \mathcal{F}_3(\mathbf{z}^s)$, $\mathbf{p}^c = \mathcal{F}_3(\mathbf{z}^c)$, $\varphi(a, b) = \exp(a \cdot b/\tau)$, and $\tau$ is the temperature parameter. This contrastive learning scheme encourages the model to focus on inter-modal/slice contextual differences, improving anatomical coherence and fine-grained detail preservation. The overall loss of CoPeVAE is formulated as

$$\mathcal{L}_{\text{tok}} = \mathcal{L}_{\text{rec}} + \lambda(\mathcal{L}_d + \mathcal{L}_p + \mathcal{L}_c). \tag{4}$$

where $\lambda$ is the weighting coefficient, and $\mathcal{L}_{\text{rec}}$ is the reconstruction loss containing an L1 loss, vector quantization loss, adversarial loss, and perceptual loss.

Notably, within each predefined modality or slice vocabulary, the binary missing labels are used only to supervise the pretext tasks during CoPeVAE training, where missing patterns are synthetically generated from complete samples. After training, CoPeVAE is frozen and directly extracts completeness-aware prompt tokens from the observed incomplete MRI. Thus, MDiT3D is guided by learned, content-dependent prompt tokens rather than externally provided binary mask codes.

Table 1: **Quantitative results for multi-modal brain MRI synthesis on the BraTS dataset.** The numbers in the first row denote the number of missing modalities.

| | 1 | | | | | 2 | | | | | 3 | | | | |
|---|---|---|---|---|---|---|---|---|---|---|---|---|---|---|---|
| | PSNR↑ | SSIM↑ | MAE↓ | FID↓ | FVD↓ | PSNR↑ | SSIM↑ | MAE↓ | FID↓ | FVD↓ | PSNR↑ | SSIM↑ | MAE↓ | FID↓ | FVD↓ |
| *GAN-based Methods* | | | | | | | | | | | | | | | |
| MMGAN Sharma & Hamarneh (2020) | $24.71_{\pm1.57}$ | $0.817_{\pm0.027}$ | $0.089_{\pm0.023}$ | 27.94 | 489.86 | $24.38_{\pm1.74}$ | $0.806_{\pm0.023}$ | $0.093_{\pm0.029}$ | 32.48 | 726.93 | $24.06_{\pm1.92}$ | $0.794_{\pm0.031}$ | $0.102_{\pm0.035}$ | 39.37 | 898.17 |
| MMT Liu et al. (2023) | $25.19_{\pm1.41}$ | $0.824_{\pm0.017}$ | $0.089_{\pm0.027}$ | 24.53 | 527.06 | $24.50_{\pm1.55}$ | $0.811_{\pm0.020}$ | $0.092_{\pm0.025}$ | 29.57 | 686.52 | $23.92_{\pm1.53}$ | $0.801_{\pm0.021}$ | $0.099_{\pm0.038}$ | 39.66 | 841.46 |
| Hyper-GAE Yang et al. (2023) | $24.65_{\pm1.62}$ | $0.813_{\pm0.022}$ | $0.087_{\pm0.018}$ | 28.97 | 609.65 | $24.42_{\pm1.74}$ | $0.808_{\pm0.024}$ | $0.096_{\pm0.021}$ | 33.52 | 815.92 | $23.86_{\pm1.88}$ | $0.788_{\pm0.029}$ | $0.105_{\pm0.029}$ | 41.79 | 948.27 |
| *Diffusion Model-based Methods* | | | | | | | | | | | | | | | |
| LDM Rombach et al. (2022) | $23.84_{\pm1.52}$ | $0.805_{\pm0.019}$ | $0.096_{\pm0.032}$ | 36.47 | 740.26 | $23.12_{\pm1.64}$ | $0.798_{\pm0.021}$ | $0.102_{\pm0.030}$ | 45.93 | 897.83 | $22.65_{\pm1.77}$ | $0.791_{\pm0.025}$ | $0.108_{\pm0.040}$ | 52.50 | 1161.21 |
| ControlNet Zhang et al. (2023) | $23.98_{\pm1.49}$ | $0.808_{\pm0.018}$ | $0.094_{\pm0.024}$ | 34.09 | 806.82 | $23.34_{\pm1.60}$ | $0.801_{\pm0.020}$ | $0.104_{\pm0.029}$ | 41.28 | 986.30 | $22.85_{\pm1.68}$ | $0.795_{\pm0.022}$ | $0.109_{\pm0.047}$ | 48.07 | 1074.23 |
| M2DN Meng et al. (2024) | $26.45_{\pm1.36}$ | $0.830_{\pm0.016}$ | $0.077_{\pm0.031}$ | 21.29 | 376.53 | $25.87_{\pm1.48}$ | $0.820_{\pm0.017}$ | $0.082_{\pm0.024}$ | 27.36 | 467.66 | $25.21_{\pm1.59}$ | $0.809_{\pm0.024}$ | $0.093_{\pm0.022}$ | 32.40 | 553.16 |
| DiffM$^4$RI Ye et al. (2026) | $26.07_{\pm1.57}$ | $0.824_{\pm0.024}$ | $0.074_{\pm0.028}$ | 25.03 | 449.17 | $25.49_{\pm1.41}$ | $0.813_{\pm0.025}$ | $0.086_{\pm0.030}$ | 32.59 | 591.35 | $25.08_{\pm1.75}$ | $0.806_{\pm0.029}$ | $0.090_{\pm0.032}$ | 35.91 | 695.06 |
| APT Shin et al. (2025) | $26.96_{\pm1.45}$ | $0.833_{\pm0.020}$ | $0.061_{\pm0.029}$ | 19.43 | 387.79 | $26.40_{\pm1.72}$ | $0.817_{\pm0.019}$ | $0.065_{\pm0.033}$ | 24.83 | 420.38 | $25.87_{\pm1.69}$ | $0.808_{\pm0.027}$ | $0.073_{\pm0.029}$ | 28.93 | 571.29 |
| ASLDM Zhou et al. (2026) | $27.36_{\pm1.19}$ | $0.835_{\pm0.026}$ | $0.059_{\pm0.020}$ | 17.13 | 339.20 | $26.97_{\pm1.32}$ | $0.820_{\pm0.015}$ | $0.064_{\pm0.022}$ | 21.72 | 431.83 | $26.09_{\pm1.62}$ | $0.811_{\pm0.020}$ | $0.070_{\pm0.017}$ | 26.40 | 491.37 |
| **CoPeDiT** | $\mathbf{28.26_{\pm1.24}}$ | $\mathbf{0.842_{\pm0.019}}$ | $\mathbf{0.055_{\pm0.023}}$ | **12.67** | **254.71** | $\mathbf{28.13_{\pm1.49}}$ | $\mathbf{0.831_{\pm0.021}}$ | $\mathbf{0.058_{\pm0.027}}$ | **13.25** | **287.58** | $\mathbf{27.91_{\pm1.41}}$ | $\mathbf{0.822_{\pm0.023}}$ | $\mathbf{0.063_{\pm0.024}}$ | **14.89** | **323.19** |

Table 2: **Quantitative results for multi-modal brain MRI synthesis on the IXI dataset**. The numbers in the first row denote the number of missing modalities.

| | 1 | | | | | 2 | | | | |
|---|---|---|---|---|---|---|---|---|---|---|
| | PSNR↑ | SSIM↑ | MAE↓ | FID↓ | FVD↓ | PSNR↑ | SSIM↑ | MAE↓ | FID↓ | FVD↓ |
| *GAN-based Methods* | | | | | | | | | | |
| MMGAN Sharma & Hamarneh (2020) | $22.29_{\pm1.35}$ | $0.684_{\pm0.015}$ | $0.100_{\pm0.027}$ | 70.91 | 1447.83 | $21.13_{\pm1.49}$ | $0.668_{\pm0.019}$ | $0.107_{\pm0.034}$ | 93.57 | 1787.39 |
| MMT Liu et al. (2023) | $22.64_{\pm1.49}$ | $0.698_{\pm0.013}$ | $0.098_{\pm0.035}$ | 53.60 | 1329.25 | $21.82_{\pm1.60}$ | $0.687_{\pm0.019}$ | $0.102_{\pm0.029}$ | 72.24 | 1562.44 |
| Hyper-GAE Yang et al. (2023) | $22.12_{\pm1.33}$ | $0.682_{\pm0.017}$ | $0.100_{\pm0.039}$ | 72.62 | 1520.49 | $20.91_{\pm1.45}$ | $0.662_{\pm0.021}$ | $0.109_{\pm0.042}$ | 98.79 | 1712.90 |
| *Diffusion Model-based Methods* | | | | | | | | | | |
| LDM Rombach et al. (2022) | $21.36_{\pm1.28}$ | $0.679_{\pm0.016}$ | $0.103_{\pm0.025}$ | 86.62 | 1884.64 | $20.94_{\pm1.53}$ | $0.654_{\pm0.020}$ | $0.112_{\pm0.029}$ | 112.57 | 2314.39 |
| ControlNet Zhang et al. (2023) | $21.83_{\pm1.51}$ | $0.681_{\pm0.015}$ | $0.101_{\pm0.046}$ | 81.26 | 1971.40 | $21.07_{\pm1.50}$ | $0.661_{\pm0.019}$ | $0.108_{\pm0.043}$ | 103.77 | 2386.04 |
| M2DN Meng et al. (2024) | $23.47_{\pm1.43}$ | $0.715_{\pm0.014}$ | $0.093_{\pm0.034}$ | 42.52 | 845.29 | $22.81_{\pm1.56}$ | $0.702_{\pm0.018}$ | $0.097_{\pm0.039}$ | 55.64 | 1078.36 |
| DiffM$^4$RI Ye et al. (2026) | $23.71_{\pm1.26}$ | $0.720_{\pm0.014}$ | $0.088_{\pm0.042}$ | 39.87 | 715.63 | $22.87_{\pm1.68}$ | $0.707_{\pm0.022}$ | $0.097_{\pm0.045}$ | 51.93 | 964.28 |
| APT Shin et al. (2025) | $24.09_{\pm1.07}$ | $0.725_{\pm0.020}$ | $0.079_{\pm0.028}$ | 30.82 | 627.43 | $23.24_{\pm1.57}$ | $0.709_{\pm0.024}$ | $0.082_{\pm0.034}$ | 44.07 | 889.78 |
| ASLDM Zhou et al. (2026) | $23.98_{\pm1.39}$ | $0.721_{\pm0.018}$ | $0.082_{\pm0.023}$ | 30.24 | 619.27 | $23.31_{\pm1.70}$ | $0.711_{\pm0.019}$ | $0.085_{\pm0.032}$ | 46.83 | 857.60 |
| **CoPeDiT** | $\mathbf{24.34_{\pm1.21}}$ | $\mathbf{0.732_{\pm0.016}}$ | $\mathbf{0.068_{\pm0.030}}$ | **25.84** | **569.22** | $\mathbf{23.92_{\pm1.53}}$ | $\mathbf{0.721_{\pm0.020}}$ | $\mathbf{0.075_{\pm0.037}}$ | **32.53** | **718.54** |

### 3.3 Stage II: 3D MRI Diffusion Transformer

We propose MDiT3D (see Fig. 3), a task-specific diffusion transformer extending DiT Peebles & Xie (2023) for volumetric MRI synthesis. MDiT3D is conditionally guided by the available latents $\mathbf{z}^s$ alongside the concatenated completeness-aware prompt tokens $\mathbf{p} = \mathbf{p}^d \,\|\, \mathbf{p}^p \,\|\, \mathbf{p}^s$. During inference, these prompt tokens are autonomously extracted by the frozen CoPeVAE to provide semantic guidance: $\mathbf{p}^d$ calibrates global severity (how many), $\mathbf{p}^p$ localizes missing regions (where), and $\mathbf{p}^s$ supplies fine-grained textural priors (what). To effectively adapt to 3D medical data, we introduce customized operations (using MDiT3D-B as a representative example). Specifically, we apply a 3D patchify operator to project inputs into tokens $\mathbf{h} \in \mathbb{R}^{m \times T \times d}$ Mo et al. (2023), where $m$ is the number of modalities/slices and $d$ is the embedding dimension. Unlike standard 2D implementations Dosovitskiy et al. (2020), we incorporate 3D frequency-based sine-cosine positional embeddings (3D PE) to preserve precise spatial relationships.

**Alternating Blocks & Prompt Token Injection.** We design two distinct alternating block architectures tailored to the specific characteristics of brain and cardiac MRIs. For the multi-modal brain task, we alternate between Spatial Blocks (to capture 3D spatial context) and Modal Blocks (to model inter-modal relationships). For the volumetric cardiac task, we alternate between Planar Blocks (for intra-slice features) and Spatial Blocks (for through-plane continuity). To compute attention along the appropriate axes, the latent tokens are dynamically reshaped before entering each block (e.g., aligning the modal dimension to facilitate inter-modal interaction, and then reshaped back for spatial processing). Furthermore, to maximize the efficacy of conditional guidance, prompt tokens are injected via adaptive layer normalization (adaLN) Peebles & Xie (2023) exclusively into the blocks that explicitly model the task's primary dependency. Specifically, prompt tokens are injected only into the Modal Blocks for brain MRI (addressing missing modalities) and the Spatial Blocks for cardiac MRI (addressing missing slices). This targeted injection avoids overwhelming the network and ensures the conditioning signals are both physically meaningful and highly informative.

**Joint Reconstruction & Synthesis.** During diffusion, rather than filling missing modalities or slices with zeros or tokens, we only add noise to the missing sections, keeping the available latents unperturbed to

Table 3: **Quantitative results for Cardiac MRI synthesis on the UKBB dataset.** The numbers in the first row denote the length of missing slices.

| | 8 | | | | | 16 | | | | | 24 | | | | |
|---|---|---|---|---|---|---|---|---|---|---|---|---|---|---|---|
| | PSNR↑ | SSIM↑ | MAE↓ | FID↓ | FVD↓ | PSNR↑ | SSIM↑ | MAE↓ | FID↓ | FVD↓ | PSNR↑ | SSIM↑ | MAE↓ | FID↓ | FVD↓ |
| *GAN-based Methods* | | | | | | | | | | | | | | | |
| MMGAN Sharma & Hamarneh (2020) | 25.81±0.86 | 0.815±0.012 | 0.076±0.012 | 16.68 | 392.48 | 24.35±0.84 | 0.793±0.009 | 0.080±0.013 | 27.52 | 604.37 | 23.06±1.08 | 0.776±0.018 | 0.091±0.016 | 45.88 | 767.92 |
| MMT Liu et al. (2023) | 26.02±0.82 | 0.824±0.009 | 0.075±0.016 | 15.90 | 429.72 | 24.73±0.87 | 0.809±0.010 | 0.080±0.015 | 23.75 | 570.29 | 24.12±1.06 | 0.794±0.017 | 0.086±0.015 | 37.39 | 708.68 |
| Hyper-GAE Yang et al. (2023) | 25.23±0.89 | 0.810±0.012 | 0.078±0.014 | 19.02 | 517.55 | 23.66±0.94 | 0.789±0.013 | 0.086±0.015 | 31.84 | 648.35 | 22.70±1.14 | 0.771±0.019 | 0.094±0.019 | 48.27 | 845.14 |
| *Diffusion Model-based Methods* | | | | | | | | | | | | | | | |
| LDM Rombach et al. (2022) | 24.17±0.96 | 0.795±0.012 | 0.085±0.016 | 24.61 | 704.91 | 23.04±1.03 | 0.778±0.010 | 0.093±0.016 | 44.93 | 830.73 | 22.19±1.21 | 0.761±0.018 | 0.098±0.014 | 60.02 | 910.15 |
| ControlNet Zhang et al. (2023) | 24.62±0.91 | 0.801±0.012 | 0.082±0.011 | 22.47 | 698.63 | 23.30±0.98 | 0.784±0.014 | 0.089±0.015 | 37.51 | 821.40 | 22.46±1.16 | 0.765±0.020 | 0.095±0.012 | 54.93 | 922.06 |
| M2DN Meng et al. (2024) | 25.48±0.79 | 0.814±0.011 | 0.079±0.015 | 17.72 | 453.03 | 24.62±0.87 | 0.803±0.011 | 0.082±0.016 | 24.15 | 597.78 | 24.03±0.99 | 0.780±0.021 | 0.088±0.019 | 40.08 | 823.70 |
| DiffM⁴RI Ye et al. (2026) | 25.19±0.89 | 0.808±0.014 | 0.080±0.013 | 19.16 | 495.23 | 24.78±0.80 | 0.807±0.015 | 0.083±0.013 | 22.96 | 548.44 | 24.27±1.06 | 0.791±0.018 | 0.087±0.016 | 35.27 | 674.12 |
| APT Shin et al. (2025) | 26.14±1.03 | 0.819±0.016 | 0.074±0.015 | 19.05 | 428.13 | 24.95±0.89 | 0.811±0.014 | 0.078±0.012 | 24.54 | 509.26 | 24.20±1.19 | 0.794±0.022 | 0.085±0.017 | 39.80 | 638.81 |
| ASLDM Zhou et al. (2026) | 25.34±0.87 | 0.812±0.015 | 0.077±0.013 | 18.16 | 397.53 | 24.16±0.78 | 0.801±0.014 | 0.083±0.015 | 28.79 | 622.96 | 23.12±0.98 | 0.782±0.020 | 0.090±0.014 | 47.32 | 710.67 |
| **CoPeDiT** | **26.42±0.81** | **0.831±0.013** | **0.072±0.016** | **15.53** | **318.62** | **26.07±0.74** | **0.826±0.013** | **0.074±0.014** | **18.21** | **382.04** | **25.39±0.86** | **0.817±0.016** | **0.078±0.013** | **25.84** | **490.57** |

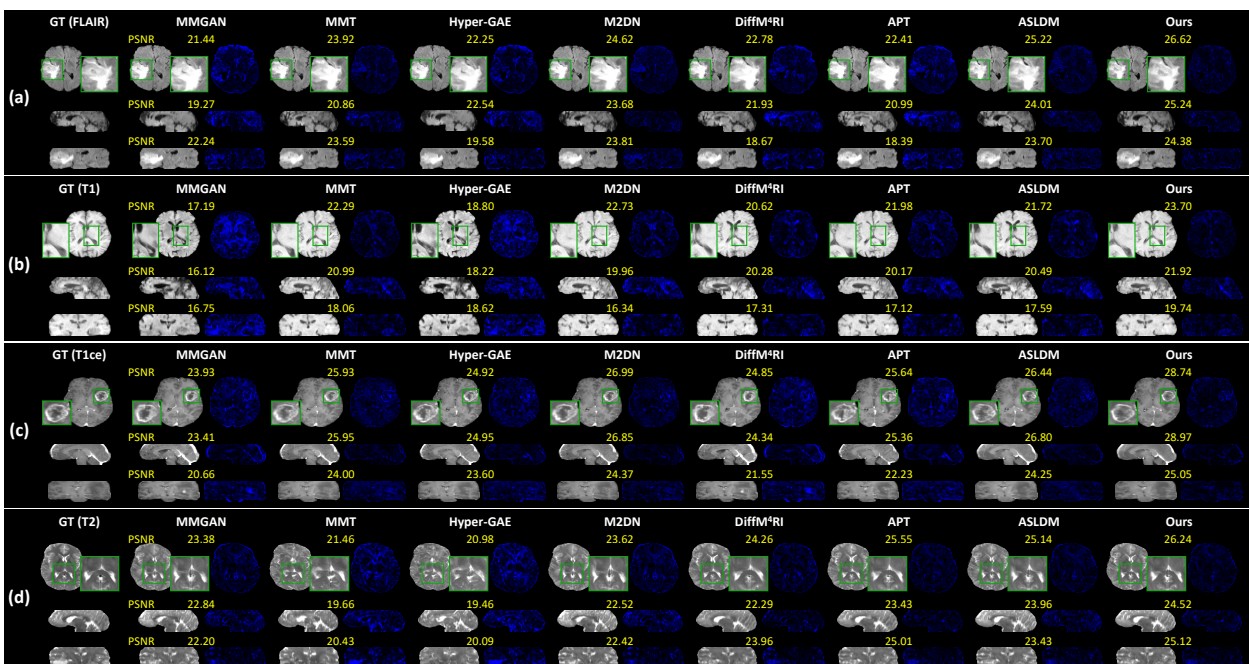

Figure 4: **Qualitative comparison on the BraTS dataset.** Rows (a)–(d) correspond to synthesizing FLAIR, T1, T1ce, and T2, respectively, from the missing modalities shown in the first column. For each case, we show three views, zoomed-in regions marked by green boxes, and the corresponding error maps.

provide contextual guidance. Following Meng et al. (2024), MDiT3D is optimized via an $\mathbf{x}_0$-prediction loss:

$$\mathcal{L}_{\text{diff}} = \mathbb{E}_{\mathbf{z}_0, \epsilon, t} \left[ \| \mathbf{z}_0 - f_\theta(\mathbf{z}_t, t, \mathbf{p}) \|_2^2 \right],$$ (5)

where $\mathbf{z}_0 = \mathbf{z}^s \| \mathbf{z}^c$ is the clean target, $\mathbf{z}_t = \mathbf{z}^s \| \mathbf{z}_t^c$ is the partially noised input at timestep $t$, $\mathbf{p}$ represents the prompt tokens, and $f_\theta$ denotes MDiT3D.

# 4 Experiments and Results

## 4.1 Experimental Setup

**Datasets. (i) Brain MRI Datasets.** We evaluate on two public brain MRI datasets: BraTS 2021 Baid et al. (2021) and IXI Brain Development Project (2025). The BraTS 2021 dataset includes 1251 subjects with multi-modal MRI scans across four modalities: T1, T1ce, T2, and FLAIR. The IXI dataset contains 577 subjects with three MRI modalities: T1, T2, and PD. **(ii) Cardiac MRI Datasets.** Missing slice synthesis experiments are conducted on four cardiac MRI datasets: UK Biobank (UKBB) Petersen et al.

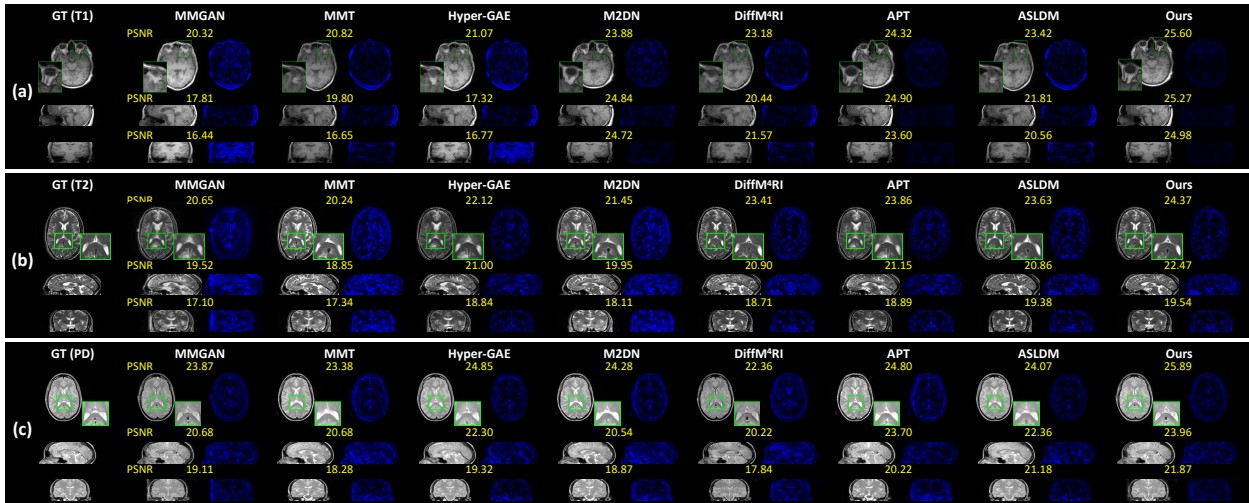

Figure 5: **Qualitative comparison on the IXI dataset.** Rows (a)–(c) correspond to synthesizing T1, T2, and PD, respectively, from the missing modalities shown in the first column.

Table 4: Ablation study on the contribution of **completeness-aware prompt tokens**.

| | BraTS | | | | | | | | | | | | | | | UKBB | | | | | | | | | |
| | 1 | | | | | 2 | | | | | 3 | | | | | 8 | | | | | 24 | | | | |
| | PSNR↑ | SSIM↑ | MAE↓ | FID↓ | FVD↓ | PSNR↑ | SSIM↑ | MAE↓ | FID↓ | FVD↓ | PSNR↑ | SSIM↑ | MAE↓ | FID↓ | FVD↓ | PSNR↑ | SSIM↑ | MAE↓ | FID↓ | FVD↓ | PSNR↑ | SSIM↑ | MAE↓ | FID↓ | FVD↓ |
|---|---|---|---|---|---|---|---|---|---|---|---|---|---|---|---|---|---|---|---|---|---|---|---|---|---|
| w/o $\mathbf{p}^d$ | 27.35 | 0.833 | 0.060 | 16.04 | 335.08 | 27.02 | 0.824 | 0.064 | 17.30 | 475.32 | 26.73 | 0.814 | 0.069 | 19.62 | 556.79 | 25.82 | 0.819 | 0.076 | 16.34 | 427.06 | 24.53 | 0.803 | 0.085 | 33.80 | 610.46 |
| w/o $\mathbf{p}^p$ | 26.92 | 0.829 | 0.063 | 18.46 | 372.36 | 26.43 | 0.819 | 0.067 | 20.23 | 460.45 | 26.06 | 0.810 | 0.073 | 25.13 | 572.03 | 25.17 | 0.812 | 0.078 | 20.08 | 534.80 | 24.39 | 0.798 | 0.086 | 36.54 | 697.51 |
| w/o $\mathbf{p}^s$ | 27.56 | 0.835 | 0.058 | 15.26 | 341.70 | 27.22 | 0.827 | 0.060 | 16.37 | 395.43 | 26.90 | 0.815 | 0.065 | 17.58 | 468.23 | 26.27 | 0.828 | 0.074 | 16.79 | 388.46 | 25.08 | 0.813 | 0.080 | 29.83 | 578.13 |
| w/o Prompt Tokens | 25.92 | 0.823 | 0.069 | 25.69 | 418.12 | 25.06 | 0.807 | 0.080 | 32.17 | 556.92 | 24.83 | 0.802 | 0.089 | 37.97 | 748.25 | 24.70 | 0.797 | 0.083 | 23.29 | 721.35 | 23.56 | 0.778 | 0.090 | 42.17 | 830.72 |
| w/ Mask Codes | 27.18 | 0.831 | 0.060 | 17.42 | 356.09 | 26.82 | 0.816 | 0.063 | 20.06 | 439.13 | 26.18 | 0.809 | 0.071 | 24.59 | 543.82 | 26.15 | 0.823 | 0.074 | 18.15 | 409.47 | 24.65 | 0.802 | 0.081 | 35.86 | 612.73 |
| $\mathbf{p}^p \to$ Mask Codes | 27.69 | 0.837 | 0.057 | 15.83 | 328.09 | 27.14 | 0.825 | 0.061 | 18.26 | 437.53 | 26.76 | 0.815 | 0.068 | 22.14 | 552.57 | 25.99 | 0.820 | 0.075 | 16.57 | 441.29 | 24.72 | 0.801 | 0.084 | 33.42 | 589.32 |
| **CoPeDiT** | **28.26** | **0.842** | **0.055** | **12.67** | **254.71** | **28.13** | **0.831** | **0.058** | **13.25** | **287.58** | **27.91** | **0.822** | **0.063** | **14.89** | **323.19** | **26.42** | **0.831** | **0.072** | **15.53** | **318.62** | **25.39** | **0.817** | **0.078** | **25.84** | **490.57** |

(2016), MESA Zhang et al. (2018), ACDC Bernard et al. (2018), and MSCMR Zhuang et al. (2022). The model is trained on the combined dataset including all four sources with 32,248 MRI volumes in total, while performance comparisons are conducted on the UKBB dataset. We randomly select 80% of the data for training and use the remaining 20% as the test set. Please refer to Appendix A for more details.

**Implementation Details.** The compression rate of CoPeVAE is set to $(8, 8, 8)$. The dimension of each prompt token is set to 512, $\tau$ and $\lambda$ are set to 0.2 and `1e-2`, respectively. The model is trained with a global batch size of 8 for CoPeVAE-B and 64 for CoPeVAE-C, using a learning rate of `1e-4`. Regarding MDiT3D, we set the number of time steps to 500 with linearly scaled noise scheduling. The model is trained for 100k steps with a global batch size of 32 for MDiT3D-B and 64 for MDiT3D-C, with learning rate of `5e-5`. All training is conducted on four NVIDIA A100 GPUs. Brain and cardiac experiments are trained as separate task-specific instantiations, while sharing the same completeness-perception principle, prompt token construction scheme, and diffusion-based synthesis pipeline. More details are provided in Appendix B.

## 4.2 Performance Comparison

We compare our method against nine SOTA baselines, including three GAN-based approaches Sharma & Hamarneh (2020); Liu et al. (2023); Yang et al. (2023) and six diffusion-based models Rombach et al. (2022); Zhang et al. (2023); Meng et al. (2024); Ye et al. (2026); Shin et al. (2025); Zhou et al. (2026), all reimplemented under identical settings for fair comparison. Performance is evaluated using Peak Signal-to-Noise Ratio (PSNR), Structural Similarity Index Measure (SSIM), Mean Absolute Error (MAE), Fréchet Inception Distance (FID) Heusel et al. (2017), and Fréchet Video Distance (FVD) Unterthiner et al. (2018) for assessing 3D spatial consistency. We also provide lesion-wise and anatomical metric evaluation in Appendix C.1.

**Quantitative Results.** Quantitative results on the three datasets are presented in Tables 1, 2 and 3, respectively. CoPeDiT consistently outperforms all baselines across all missing configurations in both synthesis

Table 5: Quantitative results by **incorporating our learned prompt tokens into baselines** instead of mask codes.

| | PSNR↑ | SSIM↑ | MAE↓ | FID↓ | FVD↓ |
|---|---|---|---|---|---|
| MMT Liu et al. (2023) | 25.19 | 0.824 | 0.089 | 24.53 | 527.06 |
| + **Prompt Tokens** (ours) | 25.68 (+0.49) | 0.826 (+0.002) | 0.079 (-0.010) | 22.07 (-2.46) | 416.34 (-110.72) |
| Hyper-GAE Yang et al. (2023) | 24.65 | 0.813 | 0.087 | 28.97 | 609.65 |
| + **Prompt Tokens** (ours) | 25.26 (+0.61) | 0.822 (+0.009) | 0.080 (-0.007) | 24.26 (-4.71) | 523.75 (-85.90) |
| M2DN Meng et al. (2024) | 26.45 | 0.830 | 0.077 | 21.29 | 376.53 |
| + **Prompt Tokens** (ours) | 27.23 (+0.78) | 0.837 (+0.007) | 0.065 (-0.012) | 17.06 (-4.23) | 308.68 (-67.85) |

Table 6: Ablation study of each pretext task's contribution to **CoPeVAE's reconstruction capacity**.

| | BraTS | | IXI | | Cardiac | |
|---|---|---|---|---|---|---|
| | PSNR↑ | SSIM↑ | PSNR↑ | SSIM↑ | PSNR↑ | SSIM↑ |
| w/o Task 1 | 34.38 | 0.926 | 30.92 | 0.914 | 32.34 | 0.921 |
| w/o Task 2 | 33.62 | 0.918 | 30.35 | 0.908 | 31.59 | 0.914 |
| w/o Task 3 | 34.69 | 0.929 | 31.13 | 0.917 | 32.62 | 0.925 |
| **CoPeVAE** | **35.05** | **0.935** | **31.28** | **0.921** | **33.34** | **0.931** |

tasks. Notably, these performance gains widen in scenarios with a higher number of missing modalities or slices, such as maintaining a high 27.91 PSNR even with three missing modalities. This highlights the robustness of our completeness-aware prompt tokens in complex cases. Moreover, CoPeDiT achieves substantially lower FID and FVD scores (e.g., an FVD of 490.57 for 24 missing cardiac slices). This indicates that the generated MRIs are not only anatomically coherent and texture-preserving, but also exhibit superior 3D spatial consistency and structural continuity, enhancing perceptual realism and diagnostic plausibility. Additional multi-seed robustness analysis and unseen missing modality experiments are presented in Appendices C.2 and C.5, respectively.

**Qualitative Results.** As depicted in Figs. 4 and 5, our model generates MRIs that exhibit the highest visual similarity to the ground truth images, particularly in accurately capturing tumor regions. Our CoPeDiT excels at preserving subtle textural details and modeling anatomical structures within brain tissues, justifying our motivation that incorporating completeness perception leads to improved anatomical consistency and realism. Further results and failure case analysis are presented in Appendix C.3 and C.7, respectively.

### 4.3 Ablation Study

**Effect of Prompt Tokens.** Table 4 compares prompt tokens against one-hot mask codes. Under identical injection strategies, all individual prompt tokens and their combinations consistently outperform binary masks, providing empirical support for our completeness perception design. The positioning prompt token ($p^p$) proves most effective; its removal triggers the sharpest performance drop, likely because explicitly locating missing sections heightens sensitivity to subtle structural variations. To further explore the potential of our prompt tokens, we incorporate them into prior baselines as replacements for mask codes. As illustrated in Table 5, this substitution consistently boosts their performance, demonstrating the strong plug-and-play utility and generalizability of our method.

**Effect of Pretext Tasks.** In Table 6, CoPeVAE preserves strong reconstruction capability while benefiting from pretext tasks. Each task contributes positively, and their combination leads to further improvements. This improvement can be attributed to the fact that pretext tasks promote the tokenizer to capture anatomical structure variances in both coarse- and fine-grained manners, learning highly discriminative features. Please refer to Appendix C.6 for further evaluation on our completeness perception accuracy.

### 4.4 Tumor Segmentation

To evaluate clinical utility, we conduct downstream tumor segmentation on the BraTS dataset. Following Liu et al. (2023); Meng et al. (2024), we train a multi-modal U-Net Ronneberger et al. (2015); Isensee et al. (2021) using three available modalities alongside one modality synthesized by CoPeDiT or baselines. We also evaluate a "Missing" baseline trained solely on the three available modalities. As shown in Table 7, all synthesis methods improve upon the "Missing" baseline in terms of average Dice scores for whole tumor (WT), tumor core (TC), and enhancing tumor (ET). Notably, CoPeDiT consistently outperforms competing methods across all subregions, achieving the highest average Dice of 90.23%. These results confirm that our synthesized MRIs provide highly informative and clinically valuable inputs for downstream tasks. We further evaluate a more challenging setting where only one real modality is available and the remaining three modalities are synthesized, with results provided in Appendix C.8.

Table 7: Results of **tumor segmentation** experiments on the BraTS dataset.

| | Dice Score (%)↑ | | | |
|---|---|---|---|---|
| | WT | TC | ET | AVG |
| Missing | 86.08 | 84.67 | 81.59 | 84.11 |
| *GAN-based Methods* | | | | |
| MMGAN Sharma & Hamarneh (2020) | 89.35 | 88.14 | 87.73 | 88.41 |
| MMT Liu et al. (2023) | 90.43 | 88.37 | 86.92 | 88.57 |
| Hyper-GAE Yang et al. (2023) | 88.72 | 86.54 | 85.37 | 86.88 |
| *Diffusion Model-based Methods* | | | | |
| LDM Rombach et al. (2022) | 87.86 | 85.91 | 84.19 | 85.99 |
| ControlNet Zhang et al. (2023) | 88.27 | 87.05 | 85.23 | 86.85 |
| M2DN Meng et al. (2024) | 91.28 | 90.09 | 88.20 | 89.86 |
| DiffM⁴RI Ye et al. (2026) | 90.04 | 89.23 | 87.68 | 88.98 |
| APT Shin et al. (2025) | 90.63 | 90.12 | 88.07 | 89.61 |
| ASLDM Zhou et al. (2026) | 90.19 | 89.57 | 87.45 | 89.07 |
| **CoPeDiT** | **91.35** | **90.41** | **88.94** | **90.23** |

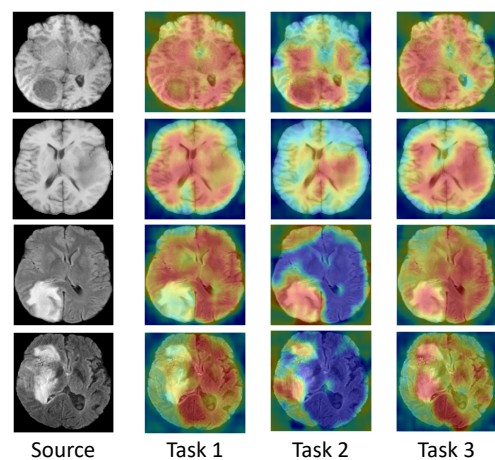

Source    Task 1    Task 2    Task 3

Figure 6: Visualization of **salient regions** on the BraTS dataset.

Table 8: The **computational cost and wall-clock time comparison** of CoPeVAE on the BraTS dataset.

| | Param. (M) | Flops (G) | Epoch Time (s) | Total Time (GPU hours) | BraTS (Avg) | |
|---|---|---|---|---|---|---|
| | | | | | PSNR↑ | FVD↓ |
| w/o Task 1 | 136.77 | 21167.61 | 331.3 | 561.2 | 27.03 | 455.73 |
| w/o Task 2 | 136.55 | 21171.83 | 332.9 | 563.9 | 26.47 | 468.13 |
| w/o Task 3 | 136.74 | 21171.83 | 330.7 | 560.1 | 27.23 | 401.79 |
| **CoPeVAE** | 137.63 | 21184.57 | 336.7 | 570.3 | **28.10** | **288.49** |

Table 9: The **inference cost** comparison of CoPeDiT on the BraTS dataset.

| | Inference Latency (s) | VRAM (G) |
|---|---|---|
| LDM Rombach et al. (2022) | 1.38 | 0.429 |
| M2DN Meng et al. (2024) | 4.67 | 0.819 |
| DiffM⁴RI Ye et al. (2026) | 2.16 | 0.584 |
| DiT-3D Mo et al. (2023) | 1.65 | 0.614 |
| **CoPeDiT** | 1.73 | 0.626 |

## 4.5 Computational Efficiency Analysis

A key concern for applying diffusion transformers to 3D medical imaging is their computational cost, since volumetric inputs contain substantially more spatial tokens than 2D images. Therefore, beyond synthesis quality, we explicitly report the computational cost of CoPeDiT under standardized settings, including model parameters, FLOPs, wall-clock training time, inference latency, and peak VRAM usage. This analysis aims to clarify whether the performance gains of CoPeDiT are obtained with substantial additional overhead, and to provide a transparent view of its accuracy–efficiency trade-off in 3D MRI synthesis.

**Training Compute and Wall-Clock Cost.** Table 8 reports the additional cost introduced by the completeness perception tasks in CoPeVAE. Compared with the variants that remove individual pretext tasks, the full CoPeVAE introduces less than 1M additional parameters and only about 0.06% extra FLOPs, while increasing the epoch time by approximately 1–2%. This small overhead leads to clear improvements in synthesis quality, increasing the average PSNR from 27.23 to 28.10 and reducing FVD from 401.79 to 288.49 compared with corresponding variants. These results indicate that the proposed pretext tasks provide a favorable accuracy–efficiency trade-off, rather than increasing complexity substantially. For the diffusion stage, MDiT3D is the main computational component, as expected for 3D diffusion transformers. Nevertheless, its complexity remains practical due to latent-space modeling and task-specific token reshaping, with the detailed model size and FLOPs reported in Appendix B.3.

**Inference Cost.** At inference time, the prompt token extraction is performed by the frozen CoPeVAE only once, while the dominant cost comes from the iterative DDIM denoising process. Table 9 reports end-to-end latency and VRAM usage per volume under standardized settings, including batch size 1, a single A100 GPU, mixed precision, and 200 DDIM sampling steps. CoPeDiT requires 1.73s and 0.626G VRAM per BraTS volume, which is close to the vanilla DiT-3D baseline (1.65s and 0.614G), while achieving substantially better synthesis quality. It is also considerably faster than heavier conditional diffusion baselines such as

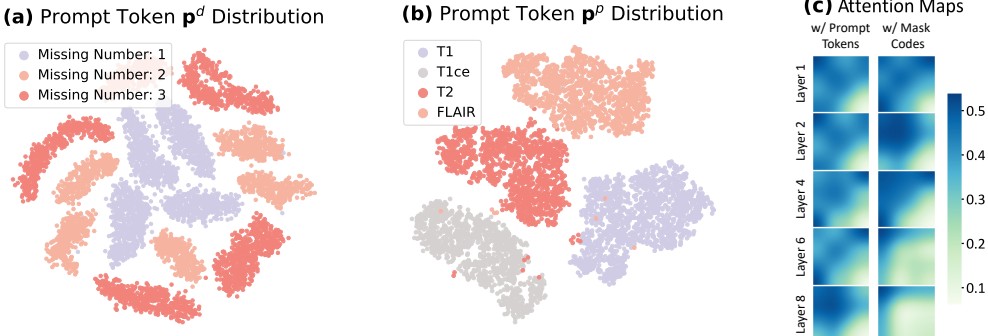

Figure 7: **Qualitative analysis of the learned prompt tokens** on the BraTS dataset. **(a)** t-SNE projection of the count-focused prompt token $\mathbf{p}^d$, showing clear clustering by the number of missing modalities. **(b)** t-SNE projection of the identity-focused prompt token $\mathbf{p}^p$, revealing distinct separation by missing modality types. **(c)** Visualization of modal block attention maps under a missing configuration of $[1, 0, 0, 0]$.

M2DN and DiffM4RI. Although CoPeDiT is slightly slower than the simplest LDM baseline, the additional cost is modest and is compensated by large gains in generation performance. Overall, CoPeDiT improves synthesis performance without incurring prohibitive computational overhead.

### 4.6 Visualization Analysis

**Salient Regions.** To understand the learning procedure of pretext tasks, we visualize their activation maps using GradCAM Selvaraju et al. (2017). As shown in Fig. 6, the examples reveal strong correlations between salient regions and modality-discriminative features. These patterns mirror each task's objective: Task 1 and Task 3 must assess global consistency, so they rely on coarse anatomical layout (gray and white matter) that summarizes the volume. In contrast, Task 2 needs to pinpoint which elements are missing and where they are located, so it keys on high-frequency, modality-specific cues (tumors, lesions, and white matter hyperintensities in FLAIR). The results emphasize that our pretext tasks capture modality-specific properties and improve the model's understanding of MRI context and inter-modality relationships.

**Prompt Token Distribution.** We implement the t-SNE visualization of learned prompt tokens and color them by ground-truth incompleteness labels. As depicted in Figs. 7a and 7b, the count-focused prompt token $\mathbf{p}^d$ forms compact, well-separated clusters aligned with the missing number classes corresponding to each missing state (1/2/3 absent modalities). Meanwhile, the identity-focused prompt token $\mathbf{p}^p$ produces modality-specific clusters (T1, T1ce, T2, FLAIR) with distinct boundaries. This clear separation visually confirms that our pretext tasks effectively learn highly discriminative features. Together, they demonstrate that our learned prompt tokens capture explicitly decoupled and complementary semantic priors. By avoiding feature entanglement, these prompt tokens successfully provide the network with nuanced, multi-granular guidance for reliable synthesis.

**Attention Maps.** We visualize the attention maps of each modal block under a specific missing configuration (e.g., $[1, 0, 0, 0]$). As shown in Fig. 7c, our learned prompt tokens actively guide the attention mechanism to progressively focus on the actual missing elements (the first row and column) as the block depth increases. This concentrated focus facilitates precise, layer-wise feature aggregation for the absent modality. In comparison, explicit mask codes lack sufficient informativeness, yielding diffuse attention weights that fail to provide the nuanced, dynamic guidance necessary to align with the true missing state.

### 4.7 Analyzing Completeness-Aware Prompt Tokens

**Prompt Token Accuracy Sensitivity.** To assess CoPeDiT's reliance on prompt token accuracy, we randomly replace the degree ($\mathbf{p}^d$), position ($\mathbf{p}^p$), semantic ($\mathbf{p}^s$), or "All" prompt tokens with incorrect ones

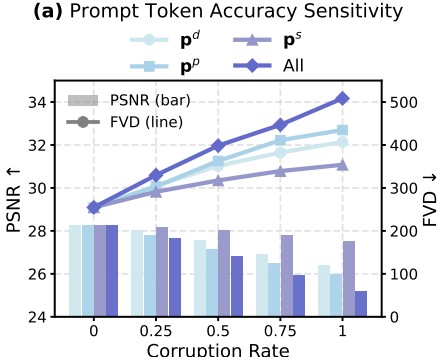 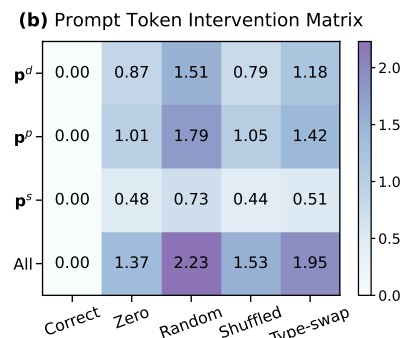

Figure 8: **Quantitative analysis of completeness-aware prompt tokens** on the BraTS dataset. **(a)** Sensitivity of CoPeDiT to increasing prompt token corruption rates. Synthesis quality degrades monotonically, particularly when perturbing the position prompt token $\mathbf{p}^p$. **(b)** Prompt token intervention matrix illustrating the PSNR drop under structured perturbations. Misleading signals (e.g., random, type-swap) and the simultaneous perturbation of all prompt tokens cause the most severe performance degradation.

for an $r$-fraction ($r \in [0, 1.0]$) of BraTS validation samples. Fig. 8a demonstrates that synthesis quality degrades consistently as the corruption rate $r$ increases. Corrupting all prompt tokens simultaneously yields the worst outcomes, confirming their complementary and essential roles.

**Prompt Token Intervention.** We apply structured interventions by replacing prompt token with zeroed, random, shuffled, or type-swapped variants. As Fig. 8b illustrates, misleading signals (random/type-swapped) degrade performance more severely than missing or mildly perturbed guidance (zeroed/shuffled). Most notably, across both sensitivity and intervention experiments, perturbing $\mathbf{p}^p$ consistently triggers the sharpest performance drops. This combined evidence robustly demonstrates that precise missing region localization ($\mathbf{p}^p$) is the most crucial prompt token for guiding high-fidelity 3D MRI synthesis.

## 5    Conclusion

This work presents CoPeDiT, a unified completeness-perception framework for 3D MRI synthesis, implemented through task-specific variants for different missing-data scenarios. We demonstrate that enabling the model to autonomously infer the missing state, rather than relying on externally predefined masks, can provide more discriminative and informative guidance. To this end, we equip our tokenizer with completeness perception capability through carefully designed pretext tasks. MDiT3D is then developed to utilize the learned prompt tokens as guidance for 3D MRI generation. Extensive evaluations validate CoPeDiT's accuracy and robustness across diverse scenarios, highlighting its potential for practical clinical deployment.

**Limitations.** While highly effective, CoPeDiT requires a fixed number of modalities during training and may lose some fine high-frequency details due to latent space compression. Future work will focus on modality-agnostic tokenizers and exploring pixel-space diffusion refinement. Moreover, as synthetic MRIs may contain hallucinated or imperfect anatomical details, they should be used only as auxiliary data and require expert validation before any clinical use.

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

## Appendix

## A    Datasets

The details of the brain and cardiac MRI datasets used in our experiments are summarized in Table 10. Notably, we train our CoPeVAE and MDiT3D models on the brain MRI synthesis task on BraTS and IXI datasets separately, due to differences in the number and types of modalities. The evaluation and results are also reported for the two datasets separately. For Cardiac MRI synthesis, we leverage a combination of all four datasets to train both stages of the model.

Table 10: Details of brain and cardiac MRI datasets.

| Datasets | Modality | Cases | Train | Test |
|---|---|---|---|---|
| *Brain MRI* | | | | |
| BraTS Baid et al. (2021) | T1, T1ce, T2, FLAIR | 1251 | 1000 | 251 |
| IXI Brain Development Project (2025) | T1, T2, PD | 577 | 462 | 115 |
| *Cardiac MRI* | | | | |
| UKBB Petersen et al. (2016) | - | 31350 | 25080 | 6270 |
| MESA Zhang et al. (2018) | - | 298 | 238 | 60 |
| ACDC Bernard et al. (2018) | - | 300 | 240 | 60 |
| MSCMR Zhuang et al. (2022) | - | 300 | 240 | 60 |
| Combined | - | 32248 | 25798 | 6450 |

## B    More Implementation Details

### B.1    Data Preprocessing

**Brain MRI data.** Following Liu et al. (2023); Meng et al. (2024), we use 90 and 80 middle axial slices for BraTS and IXI datasets, respectively. These slices are further cropped to a size of $192 \times 192$ from the central region. Ultimately, all volumes are resized to a fixed size of $192 \times 192 \times 64$ to serve as model input.

**Cardiac MRI data.** All slices within each cardiac MRI volume are used and cropped to $192 \times 192$ from the central region. Each volume is then resized to a fixed size of $192 \times 192 \times 32$ for training and inference.

For all datasets, we apply intensity normalization by linearly scaling voxel intensities between the 0.5th and 99.5th percentiles to the range $[0, 1]$. The data augmentations we employ include random spatial cropping, rotation, flipping, scaling, and shifting.

### B.2    Architecture of Prompt Token Encoders and Projection Heads

We devise three lightweight prompt token encoders to generate completeness-aware prompt tokens in our CoPeVAE, followed by three projection heads for each pretext task. The detailed architecture of each prompt token encoder and projection head is illustrated in Table 11.

Notably, in our framework, binary mask codes are only used in the pretraining of CoPeVAE, where we synthetically remove modalities/slices and supervise the pretext tasks with known missing patterns. Once CoPeVAE is trained, we freeze its parameters and use it as a completeness-aware tokenizer: given any incomplete MRI with an arbitrary missing pattern, CoPeVAE directly infers the corresponding completeness prompt tokens $\mathbf{p} = \mathbf{p}^d \, \| \, \mathbf{p}^p \, \| \, \mathbf{p}^s$ from the observed data, without requiring explicit mask codes as input. During both diffusion model training and inference, the diffusion backbone receives only the latent representations and these learned prompt tokens. The original binary masks that were used to generate synthetic missingness are not provided to the diffusion model. In this sense, CoPeDiT no longer depends on handcrafted or externally supplied mask codes at the generation stage, but instead relies entirely on the learned completeness prompt tokens produced by the frozen CoPeVAE.

Table 11: Detailed architecture of each prompt token encoder and projection head in the pretext task.

| Prompt token encoder | $\mathcal{F}_1$ | $\mathcal{F}_2$ | $\mathcal{F}_3$ |
|---|---|---|---|
| Architecture | 3D Conv
(in 8, out 256)
3D BatchNorm
ReLU
3D Conv
(in 256, out 512)
3D BatchNorm
ReLU
3D Adaptive Avg Pool
Linear (512, 1024)
ReLU
Linear (1024, 512) | 3D Conv
(in 8, out 256)
3D BatchNorm
ReLU
3D Conv
(in 256, out 512)
3D BatchNorm
ReLU
3D Adaptive Avg Pool
Linear (512, 1024)
ReLU
Linear (1024, 512) | 3D Conv
(in 8, out 256)
3D BatchNorm
ReLU
3D Conv
(in 256, out 512)
3D BatchNorm
ReLU
3D Adaptive Avg Pool
Linear (512, 1024)
ReLU
Linear (1024, 512) |
| Projection Head | $\mathcal{H}_1$ | $\mathcal{H}_2$ | $\mathcal{H}_3$ |
| Architecture | SiLU
Linear (512, $m-1$) | SiLU
Linear (512, $m$) | SiLU
Linear (512, 128) |

Table 12: Hyperparameter setup of CoPeVAE.

| | CoPeVAE-B | CoPeVAE-C |
|---|---|---|
| **Architecture** | | |
| Input dim. | $m \times 192 \times 192 \times 64$ | $192 \times 192 \times 32$ |
| Num. codebook | 8192 | 8192 |
| Latent dim. | 8 | 8 |
| Channels | (256, 384, 512) | (32, 64, 128) |
| Compression ratio | (8, 8, 8) | (8, 8, 8) |
| Prompt Token dim. | 512 | 512 |
| $\tau$ | 0.2 | 0.2 |
| $\lambda$ | 0.01 | 0.01 |
| **Optimization** | | |
| Batch size | 8 | 64 |
| Learning rate | 1e-4 | 1e-4 |
| Optimizer | Adam | Adam |
| $(\beta_1, \beta_2)$ | (0.9, 0.999) | (0.9, 0.999) |
| LR schedule | Warmup cosine | Warmup cosine |
| Training steps | 400k | 100k |

### B.3 Hyperparameter Setups

**CoPeVAE.** The detailed hyperparameter setup of CoPeVAE is provided in Table 12. Built upon VQ-VAE van den Oord et al. (2017) and VQGAN Esser et al. (2021), our model employs a codebook containing 8192 codes with the latent dimensionality of 8. The values of $\tau$ and $\lambda$ are empirically set, as they have only a slight impact on model performance. The Adam optimizer is applied with a warmup cosine learning rate schedule. The training steps of CoPeVAE-B and CoPeVAE-C are 400k and 100k, respectively.

**MDiT3D.** For MDiT3D, we carefully design the hyperparameters to balance dataset size and model capacity, as summarized in Table 13. Following DiT Peebles & Xie (2023), we report the experimental results using the exponential moving average (EMA) with a decay rate of 0.9999. During training, we set the timestep to 500 with linearly scaled noise levels ranging from 0.0015 to 0.0195. MDiT3D is trained for 100k iterations using the AdamW optimizer and a warmup cosine learning rate schedule. During inference, the DDIM

Table 13: Hyperparameter setup of MDiT3D.

|  | MDiT3D-B | MDiT3D-C |
|---|---|---|
| **Architecture** | | |
| Input dim. | $m \times 8 \times 24 \times 24 \times 8$ | $4 \times 8 \times 24 \times 24$ |
| Hidden dim. | 768 | 576 |
| Num. blocks | 16 | 12 |
| Num. heads | 12 | 12 |
| Patch size | 2 | 1 |
| Params (M) | 173.3 | 33.0 |
| Flops (G) | 555.1 (BraTS) 424.5 (IXI) | 104.0 |
| **Optimization** | | |
| Batch size | 32 | 64 |
| Learning rate | 5e-5 | 5e-5 |
| Optimizer | AdamW | AdamW |
| $(\beta_1, \beta_2)$ | (0.9, 0.999) | (0.9, 0.999) |
| LR schedule | Warmup cosine | Warmup cosine |
| Training steps | 100k | 100k |
| EMA decay | 0.9999 | 0.9999 |
| **Interpolants** | | |
| Training objective | $\mathbf{x}_0$-prediction | $\mathbf{x}_0$-prediction |
| Noise schedule | scaled-linear | scaled-linear |
| Timesteps | 500 | 500 |
| Sampler | DDIM | DDIM |
| Sampling steps | 200 | 250 |

Table 14: **Lesion-wise fidelity and segmentation-based anatomical consistency on the BraTS dataset.** The numbers in the first row denote the number of missing modalities. All metrics are averaged over three tumor regions, including WT, TC and ET.

|  | 1 | | | | 3 | | | |
|---|---|---|---|---|---|---|---|---|
|  | **Lesion-wise Fidelity** | | **Anatomical Consistency** | | **Lesion-wise Fidelity** | | **Anatomical Consistency** | |
|  | MAE↓ | SSIM↑ | Dice (%)↑ | HD95 (mm)↓ | MAE↓ | SSIM↑ | Dice (%)↑ | HD95 (mm)↓ |
| *GAN-based Methods* | | | | | | | | |
| MMGAN Sharma & Hamarneh (2020) | $0.095_{\pm 0.025}$ | $0.792_{\pm 0.036}$ | $83.91_{\pm 10.40}$ | $11.09_{\pm 4.57}$ | $0.108_{\pm 0.043}$ | $0.786_{\pm 0.035}$ | $81.67_{\pm 9.42}$ | $11.83_{\pm 5.62}$ |
| MMT Liu et al. (2023) | $0.092_{\pm 0.019}$ | $0.792_{\pm 0.022}$ | $83.66_{\pm 12.05}$ | $11.24_{\pm 5.08}$ | $0.098_{\pm 0.024}$ | $0.784_{\pm 0.020}$ | $81.80_{\pm 11.59}$ | $12.08_{\pm 4.38}$ |
| Hyper-GAE Yang et al. (2023) | $0.093_{\pm 0.031}$ | $0.787_{\pm 0.034}$ | $84.18_{\pm 12.43}$ | $10.74_{\pm 4.26}$ | $0.105_{\pm 0.034}$ | $0.783_{\pm 0.032}$ | $81.29_{\pm 8.96}$ | $12.27_{\pm 5.40}$ |
| *Diffusion Model-based Methods* | | | | | | | | |
| LDM Rombach et al. (2022) | $0.106_{\pm 0.036}$ | $0.784_{\pm 0.029}$ | $84.05_{\pm 8.46}$ | $10.59_{\pm 3.62}$ | $0.109_{\pm 0.041}$ | $0.780_{\pm 0.039}$ | $80.88_{\pm 12.43}$ | $13.29_{\pm 3.76}$ |
| ControlNet Zhang et al. (2023) | $0.102_{\pm 0.035}$ | $0.789_{\pm 0.042}$ | $84.23_{\pm 10.86}$ | $10.40_{\pm 4.17}$ | $0.112_{\pm 0.038}$ | $0.778_{\pm 0.027}$ | $80.59_{\pm 11.27}$ | $13.04_{\pm 3.91}$ |
| M2DN Meng et al. (2024) | $0.084_{\pm 0.028}$ | $0.801_{\pm 0.036}$ | $85.22_{\pm 9.26}$ | $9.76_{\pm 4.82}$ | $0.090_{\pm 0.031}$ | $0.792_{\pm 0.039}$ | $81.35_{\pm 8.72}$ | $12.64_{\pm 4.79}$ |
| DiffM$^4$RI Ye et al. (2026) | $0.086_{\pm 0.039}$ | $0.798_{\pm 0.040}$ | $84.80_{\pm 11.42}$ | $10.03_{\pm 4.07}$ | $0.094_{\pm 0.032}$ | $0.790_{\pm 0.032}$ | $81.56_{\pm 9.51}$ | $12.24_{\pm 5.19}$ |
| APT Shin et al. (2025) | $0.075_{\pm 0.034}$ | $0.810_{\pm 0.030}$ | $87.08_{\pm 10.33}$ | $8.29_{\pm 3.73}$ | $0.086_{\pm 0.041}$ | $0.799_{\pm 0.034}$ | $82.93_{\pm 12.04}$ | $10.85_{\pm 5.30}$ |
| ASLDM Zhou et al. (2026) | $0.073_{\pm 0.029}$ | $0.809_{\pm 0.025}$ | $86.77_{\pm 11.71}$ | $8.47_{\pm 3.80}$ | $0.086_{\pm 0.035}$ | $0.799_{\pm 0.029}$ | $83.48_{\pm 9.68}$ | $10.57_{\pm 4.47}$ |
| **CoPeDiT** | $\mathbf{0.067_{\pm 0.036}}$ | $\mathbf{0.815_{\pm 0.028}}$ | $\mathbf{87.63_{\pm 9.52}}$ | $\mathbf{7.98_{\pm 4.12}}$ | $\mathbf{0.078_{\pm 0.041}}$ | $\mathbf{0.802_{\pm 0.030}}$ | $\mathbf{85.09_{\pm 11.27}}$ | $\mathbf{9.19_{\pm 4.23}}$ |

sampler Song et al. (2021) is applied with sampling steps of 200 and 250 for MDiT3D-B and MDiT3D-C, respectively.

In addition, we use mixed-precision training with gradient clipping to accelerate training and save computational resources throughout all two-stage experiments.

Table 15: **Multi-seed robustness analysis on challenging brain MRI synthesis settings.** We report the mean and standard deviation over three random seeds.

| Method | BraTS: 3 Missing Modalities | | | | | IXI: 2 Missing Modalities | | | | |
|---|---|---|---|---|---|---|---|---|---|---|
| | PSNR↑ | SSIM↑ | MAE↓ | FID↓ | FVD↓ | PSNR↑ | SSIM↑ | MAE↓ | FID↓ | FVD↓ |
| M2DN Meng et al. (2024) | $25.15_{\pm 0.39}$ | $0.808_{\pm 0.005}$ | $0.091_{\pm 0.006}$ | $32.04_{\pm 5.11}$ | $540.42_{\pm 27.22}$ | $22.83_{\pm 0.41}$ | $0.703_{\pm 0.004}$ | $0.096_{\pm 0.003}$ | $56.11_{\pm 7.65}$ | $1060.50_{\pm 65.95}$ |
| ASLDM Zhou et al. (2026) | $26.16_{\pm 0.21}$ | $0.811_{\pm 0.004}$ | $0.070_{\pm 0.006}$ | $25.94_{\pm 3.30}$ | $498.32_{\pm 30.56}$ | $23.29_{\pm 0.42}$ | $0.712_{\pm 0.005}$ | $0.085_{\pm 0.006}$ | $46.34_{\pm 5.76}$ | $867.91_{\pm 47.79}$ |
| **CoPeDiT** | $\mathbf{27.78}_{\pm 0.28}$ | $\mathbf{0.821}_{\pm 0.005}$ | $\mathbf{0.062}_{\pm 0.004}$ | $\mathbf{15.07}_{\pm 2.63}$ | $\mathbf{316.08}_{\pm 23.46}$ | $\mathbf{23.88}_{\pm 0.36}$ | $\mathbf{0.721}_{\pm 0.006}$ | $\mathbf{0.073}_{\pm 0.005}$ | $\mathbf{32.97}_{\pm 4.56}$ | $\mathbf{729.28}_{\pm 56.44}$ |

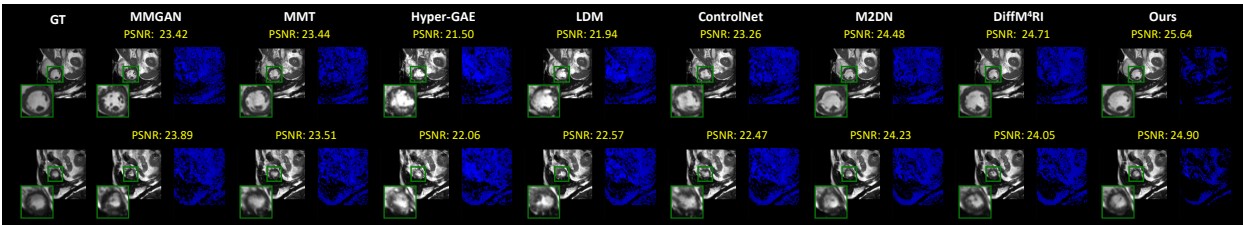

Figure 9: **Qualitative results on the UKBB cardiac MRI dataset.** The top and bottom results correspond to the first and last missing slices within a given volume, respectively. For each example, we compare synthesized results from different methods with the ground truth, together with zoomed-in regions and the corresponding error maps.

## C  Additional Experimental Results

### C.1  Lesion-wise and Anatomical Consistency Evaluation

To further evaluate pathological fidelity beyond global image similarity, we conduct additional lesion-aware evaluation on the BraTS dataset. Specifically, we use the ground-truth tumor annotations to define three clinically relevant regions, including WT, TC and ET. For lesion-wise fidelity, MAE and SSIM are computed within each lesion region between the synthesized and reference target modalities. For anatomical consistency, we use a frozen tumor segmentation model, i.e., nnU-Net Isensee et al. (2021) trained on real images, to segment the completed multi-modal MRIs, where missing modalities are replaced by synthesized ones. We report Dice coefficient (Dice) and 95th-percentile Hausdorff distance (HD95) against the ground-truth tumor labels. All metrics are averaged over test subjects, synthesized missing modalities, and the three tumor regions. As reported in Table 14, CoPeDiT achieves the best average lesion-wise fidelity and anatomical consistency under both one- and three-missing-modality settings. In particular, CoPeDiT obtains lower MAE and higher SSIM in lesion regions, while also achieving higher Dice and lower HD95 in segmentation-based evaluation. The competitive lesion-wise and anatomical consistency performances of CoPeDiT indicate that our proposed completeness-aware prompt tokens improve not only global synthesis quality but also the preservation of local pathological structures.

### C.2  Multi-seed Robustness Analysis

To provide additional statistical support under feasible computational cost, we conduct a small-scale multi-seed robustness analysis on the challenging brain MRI synthesis settings, including three missing modalities on BraTS and two missing modalities on IXI. We compare CoPeDiT with two representative diffusion-based baselines, i.e., M2DN Meng et al. (2024), and ASLDM Zhou et al. (2026), using three independent random seeds. For each method, we report the mean and standard deviation across seeds. The experimental results in Table 15 demonstrate that CoPeDiT maintains consistent advantages over the strongest baselines on both datasets, indicating that the observed improvements are not caused by a single random run. This experiment is intended as a focused robustness analysis rather than exhaustive cross-validation, due to the high computational cost of training 3D latent diffusion models on volumetric MRI data.

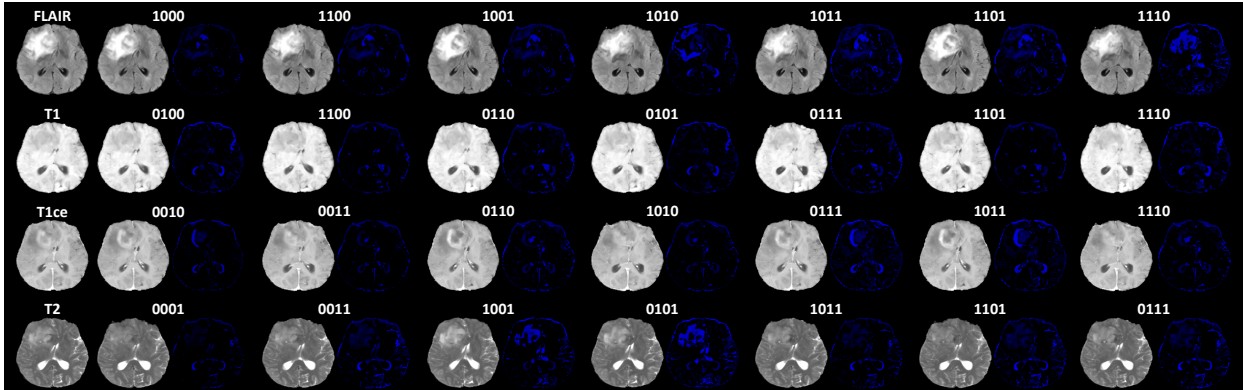

Figure 10: **Qualitative results of CoPeDiT on the BraTS dataset under all missing-modality combinations.** Each row corresponds to one target modality. The leftmost image in each row is the ground truth. The remaining columns show all missing scenarios for the target modality, denoted by a binary vector in the order of FLAIR, T1, T1ce, and T2, where 1/0 indicates missing/available modality, respectively.

Table 16: Ablation study on the contribution of completeness perception prompt tokens on the IXI dataset.

| | IXI | | | | | | | | | |
|---|---|---|---|---|---|---|---|---|---|---|
| | 1 | | | | | 2 | | | | |
| | PSNR↑ | SSIM↑ | MAE↓ | FID↓ | FVD↓ | PSNR↑ | SSIM↑ | MAE↓ | FID↓ | FVD↓ |
| w/o $\mathbf{p}^d$ | 23.56 | 0.720 | 0.089 | 35.93 | 724.18 | 23.04 | 0.708 | 0.095 | 48.71 | 917.76 |
| w/o $\mathbf{p}^p$ | 22.81 | 0.697 | 0.099 | 50.70 | 1267.70 | 21.99 | 0.692 | 0.101 | 63.16 | 1281.31 |
| w/o $\mathbf{p}^s$ | 24.17 | 0.724 | 0.082 | 31.89 | 685.13 | 23.56 | 0.710 | 0.087 | 39.87 | 876.94 |
| w/o Prompt Tokens | 22.48 | 0.696 | 0.102 | 57.64 | 1315.59 | 21.67 | 0.683 | 0.106 | 81.49 | 1542.24 |
| w/ Mask Codes | 23.35 | 0.711 | 0.093 | 44.03 | 783.46 | 22.84 | 0.702 | 0.099 | 56.72 | 1067.16 |
| $\mathbf{p}^p \rightarrow$ Mask Codes | 24.10 | 0.724 | 0.081 | 32.26 | 594.83 | 23.37 | 0.708 | 0.084 | 49.24 | 930.09 |
| **CoPeDiT** | **24.34** | **0.732** | **0.068** | **25.84** | **569.22** | **23.92** | **0.721** | **0.075** | **32.53** | **718.54** |

## C.3 Additional Qualitative Results

We further provide qualitative comparisons for cardiac MRI synthesis on the UKBB dataset in Fig. 9. For each example, we show the synthesized slice, a zoomed-in region, and the corresponding error map. Compared with competing methods, CoPeDiT produces lower local synthetic errors, which further confirms the superior structural fidelity and spatial consistency of our method in missing-slice synthesis.

To further examine the performance of CoPeDiT under diverse incomplete inputs, we visualize qualitative results for all target modalities and all missing combinations on the BraTS and IXI datasets in Figs. 10 and 11. For each target modality, we show the synthesized image together with the corresponding error map. As observed, CoPeDiT maintains consistent anatomical structures and modality-specific contrast patterns across diverse missing scenarios. The errors remain mainly localized in fine structural or lesion-related regions and become more noticeable only in more challenging cases with fewer available input modalities.

## C.4 More Ablation Study

**Effect of Prompt Tokens on IXI.** We further evaluate the effectiveness of our prompt token design on the IXI dataset. As shown in Table 16, the complete set of prompt tokens yields the best performance, outperforming both conventional mask codes and all individual prompt tokens. Furthermore, we apply our learned prompt tokens to other baseline models originally using mask codes. As illustrated in Table 17, our prompt tokens also lead to consistent performance gains across all baselines. In summary, the effectiveness of our prompt token learning scheme is validated on the IXI dataset through the additional evaluations presented above.

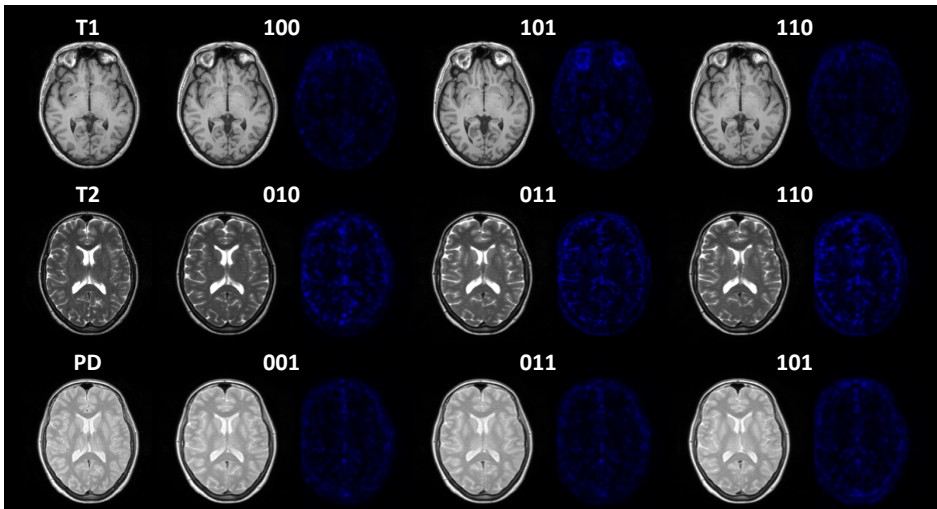

Figure 11: **Qualitative results of CoPeDiT on the IXI dataset under all missing-modality combinations.** Each row corresponds to one target modality. The leftmost image in each row is the ground truth. The remaining columns show all missing scenarios for the target modality, denoted by a binary vector in the order of T1, T2, and PD, where 1/0 indicates missing/available modality, respectively.

Table 17: Quantitative results on the IXI dataset by incorporating our completeness perception prompt tokens into baseline methods instead of mask codes.

|  | PSNR↑ | SSIM↑ | MAE↓ | FID↓ | FVD↓ |
|---|---|---|---|---|---|
| MMT Liu et al. (2023) | 22.64 | 0.698 | 0.098 | 53.60 | 1329.25 |
| + **Prompt Tokens** (ours) | 23.19 (+0.55) | 0.707 (+0.009) | 0.092 (-0.006) | 46.13 (−7.47) | 1096.06 (−233.19) |
| Hyper-GAE Yang et al. (2023) | 22.12 | 0.682 | 0.100 | 72.62 | 1520.49 |
| + **Prompt Tokens** (ours) | 22.46 (+0.34) | 0.694 (+0.012) | 0.093 (-0.007) | 59.43 (−13.19) | 1261.53 (−258.96) |
| M2DN Meng et al. (2024) | 23.47 | 0.715 | 0.093 | 42.52 | 845.29 |
| + **Prompt Tokens** (ours) | 23.84 (+0.37) | 0.726 (+0.011) | 0.083 (-0.010) | 33.79 (−8.73) | 684.62 (−160.67) |

**Choice of Prompt Token Injection Position.** We aim to justify our choice of injecting prompt tokens exclusively into the modal and spatial blocks for brain and cardiac tasks, respectively. As shown in Table 18, injecting prompt tokens into the modal/spatial block yields the best performance, whereas injecting into spatial/planar blocks or both blocks results in inferior outcomes. This can be attributed to the fact that aligning the prompt tokens with the block that explicitly models the task's primary dependency, namely modality fusion for brain and through-plane continuity for cardiac, maximizes the efficacy of conditional signals.

**Impact of 3D MRI Diffusion Transformer.** The evaluation of MDiT3D and existing diffusers Rombach et al. (2022); Peebles & Xie (2023); Mo et al. (2023) is presented in Table 19. MDiT3D achieves superior performance, notably outperforming vanilla 3D DiTs by over 1.4 dB in PSNR on the BraTS dataset. This confirms that our task-driven design, which couples alternating blocks with targeted prompt token injection to explicitly model inter-modal relationships and through-plane continuity, effectively improves synthesis.

### C.5 Robustness to Unseen Missing Modality Combinations

To evaluate the robustness of CoPeDiT to unseen missing patterns, we conduct a leave-combination-out experiment on the BraTS dataset. Specifically, several missing modality combinations are excluded during training and used only for testing, ensuring that evaluation is performed on previously unseen incomplete configurations. We compare CoPeDiT with two variants, i.e., removing prompt token guidance and replacing learned prompt tokens with binary mask codes, under three held-out modality settings in Table 20. As shown,

Table 18: Ablation study on the choice of **prompt token injection positions within the MDiT3D blocks.** The labels denote the target blocks for brain and cardiac tasks, respectively (e.g., "Modal / Spatial" means injecting prompt tokens exclusively into the modal blocks for brain MRIs and spatial blocks for cardiac MRIs). "Both" indicates injection into both types of blocks for the respective tasks.

| | BraTS | | | | | | | | | | | | | | | UKBB | | | | | | | | | |
| | 1 | | | | | 2 | | | | | 3 | | | | | 8 | | | | | 24 | | | | |
| | PSNR↑ | SSIM↑ | MAE↓ | FID↓ | FVD↓ | PSNR↑ | SSIM↑ | MAE↓ | FID↓ | FVD↓ | PSNR↑ | SSIM↑ | MAE↓ | FID↓ | FVD↓ | PSNR↑ | SSIM↑ | MAE↓ | FID↓ | FVD↓ | PSNR↑ | SSIM↑ | MAE↓ | FID↓ | FVD↓ |
|---|---|---|---|---|---|---|---|---|---|---|---|---|---|---|---|---|---|---|---|---|---|---|---|---|---|
| Spatial / Planar | 27.47 | 0.834 | 0.060 | 15.41 | 317.62 | 27.20 | 0.825 | 0.062 | 16.83 | 484.39 | 27.03 | 0.817 | 0.068 | 17.45 | 592.17 | 26.29 | 0.827 | 0.075 | 16.82 | 405.72 | 25.22 | 0.814 | 0.080 | 29.24 | 604.32 |
| Both | 27.76 | 0.836 | 0.056 | 13.94 | 294.37 | 27.54 | 0.827 | 0.059 | 15.48 | 367.20 | 27.29 | 0.818 | 0.064 | 16.77 | 434.83 | 26.33 | 0.827 | 0.073 | 16.39 | 414.37 | 25.28 | 0.815 | 0.080 | 28.45 | 587.13 |
| **Modal / Spatial** | **28.26** | **0.842** | **0.055** | **12.67** | **254.71** | **28.13** | **0.831** | **0.058** | **13.25** | **287.58** | **27.91** | **0.822** | **0.063** | **14.89** | **323.19** | **26.42** | **0.831** | **0.072** | **15.53** | **318.62** | **25.39** | **0.817** | **0.078** | **25.84** | **490.57** |

Table 19: Ablation study on the contribution of **3D MRI Diffusion Transformer.**

| | BraTS | | | | | | | | | | | | | | | UKBB | | | | | | | | | |
| | 1 | | | | | 2 | | | | | 3 | | | | | 8 | | | | | 24 | | | | |
| | PSNR↑ | SSIM↑ | MAE↓ | FID↓ | FVD↓ | PSNR↑ | SSIM↑ | MAE↓ | FID↓ | FVD↓ | PSNR↑ | SSIM↑ | MAE↓ | FID↓ | FVD↓ | PSNR↑ | SSIM↑ | MAE↓ | FID↓ | FVD↓ | PSNR↑ | SSIM↑ | MAE↓ | FID↓ | FVD↓ |
|---|---|---|---|---|---|---|---|---|---|---|---|---|---|---|---|---|---|---|---|---|---|---|---|---|---|
| UNet Rombach et al. (2022) | 25.61 | 0.825 | 0.077 | 23.56 | 433.55 | 25.20 | 0.816 | 0.080 | 28.37 | 545.89 | 24.78 | 0.805 | 0.086 | 36.70 | 707.83 | 24.46 | 0.800 | 0.083 | 22.34 | 689.53 | 23.01 | 0.778 | 0.091 | 41.59 | 871.64 |
| DiT Peebles & Xie (2023) | 26.78 | 0.835 | 0.070 | 19.72 | 392.09 | 26.14 | 0.823 | 0.073 | 21.84 | 498.13 | 25.74 | 0.808 | 0.075 | 24.47 | 611.26 | 25.89 | 0.825 | 0.074 | 16.28 | 403.83 | 24.78 | 0.799 | 0.081 | 36.10 | 592.72 |
| DiT-3D Mo et al. (2023) | 26.84 | 0.835 | 0.069 | 19.23 | 359.83 | 26.23 | 0.822 | 0.073 | 21.95 | 487.12 | 25.90 | 0.810 | 0.078 | 26.08 | 593.42 | 26.08 | 0.825 | 0.075 | 16.37 | 397.71 | 24.83 | 0.806 | 0.081 | 33.73 | 610.89 |
| **MDiT3D** | **28.26** | **0.842** | **0.055** | **12.67** | **254.71** | **28.13** | **0.831** | **0.058** | **13.25** | **287.58** | **27.91** | **0.822** | **0.063** | **14.89** | **323.19** | **26.42** | **0.831** | **0.072** | **15.53** | **318.62** | **25.39** | **0.817** | **0.078** | **25.84** | **490.57** |

CoPeDiT consistently outperforms both variants across all unseen missing combinations. The advantage becomes more pronounced when fewer modalities are available, especially in terms of MAE, FID, and FVD, indicating better fidelity and spatial consistency under challenging missing conditions. The above unseen-missing-modality experiments suggest that our completeness-aware prompt tokens provide more transferable guidance than predefined mask codes, further supporting the robustness of CoPeDiT to previously unseen incomplete modality configurations.

## C.6 Evaluation of Completeness Perception

To validate the reliability of the prompt token encoders in CoPeVAE, we evaluate the classification performance of the pretext tasks on held-out test data. Specifically, Task 1 measures whether CoPeVAE can correctly identify the missing modality number for brain MRI or the missing slice length for cardiac MRI. Task 2 evaluates whether CoPeVAE can correctly locate the missing modalities or missing slice positions. We report accuracy and macro-F1 under each missing configuration. In Fig. 12, CoPeVAE achieves strong completeness perception across all datasets. For Task 1, the accuracy and macro-F1 remain around 98%–99% across the three datasets, confirming that CoPeVAE reliably captures the global severity of incompleteness. Task 2 is more challenging because it requires fine-grained positioning of the missing elements. Nevertheless, CoPeVAE still obtains over 90% accuracy and macro-F1 in all settings. On BraTS and IXI, the positioning performance remains stable across different missing modality numbers. On UKBB, the positioning performance increases with the missing slice length, as longer consecutive missing regions provide clearer spatial cues for localization. These evaluations indicate that our learned prompt token encoders can accurately infer both the amount and the position of missing data. Therefore, the completeness-aware prompt tokens used by our MDiT3D are grounded in reliable self-perceived missing-state representations rather than manually specified mask codes.

Moreover, to further examine how completeness perception affects downstream synthesis, we group the BraTS test cases according to the correctness of Task 1 and Task 2, and report the corresponding synthesis performance in Fig. 13. Since the two tasks are predicted by independent heads, position-correct but count-wrong cases can occur, although they are rare. Overall, the best synthesis results are consistently obtained when both the missing count and missing position are correctly inferred. Once either prediction becomes incorrect, PSNR and SSIM decrease while FVD increases, and the largest degradation is observed when both predictions are wrong. This trend is consistent across different missing-modality settings and becomes more evident as the synthesis task becomes harder. This investigation provides evidence that the quality of the learned completeness-aware prompt tokens is closely linked to the final synthesis performance.

Table 20: **Robustness to unseen missing modality combinations on the BraTS dataset.** During training, the listed missing modality combinations are excluded from both prompt token learning and diffusion training. Evaluation is conducted only on these held-out combinations.

| # Missing | Available Modalities | | | | Method | Results | | | | |
|---|---|---|---|---|---|---|---|---|---|---|
| | T1 | T1ce | T2 | FLAIR | | PSNR↑ | SSIM↑ | MAE↓ | FID↓ | FVD↓ |
| 1 | ✓ | ✓ | ✓ | | w/o Prompt Tokens | 25.49 | 0.820 | 0.070 | 29.56 | 466.08 |
| | | | | | w/ Mask Codes | 26.80 | 0.827 | 0.064 | 22.19 | 429.57 |
| | | | | | **CoPeDiT** | **28.10** | **0.839** | **0.057** | **14.19** | **302.16** |
| 2 | ✓ | | | ✓ | w/o Prompt Tokens | 24.39 | 0.804 | 0.093 | 35.59 | 724.51 |
| | | | | | w/ Mask Codes | 26.13 | 0.813 | 0.072 | 26.92 | 587.45 |
| | | | | | **CoPeDiT** | **27.92** | **0.827** | **0.059** | **15.98** | **357.83** |
| 3 | | | ✓ | | w/o Prompt Tokens | 23.50 | 0.796 | 0.103 | 46.73 | 914.50 |
| | | | | | w/ Mask Codes | 25.42 | 0.810 | 0.090 | 37.19 | 703.54 |
| | | | | | **CoPeDiT** | **27.20** | **0.817** | **0.065** | **18.75** | **458.63** |

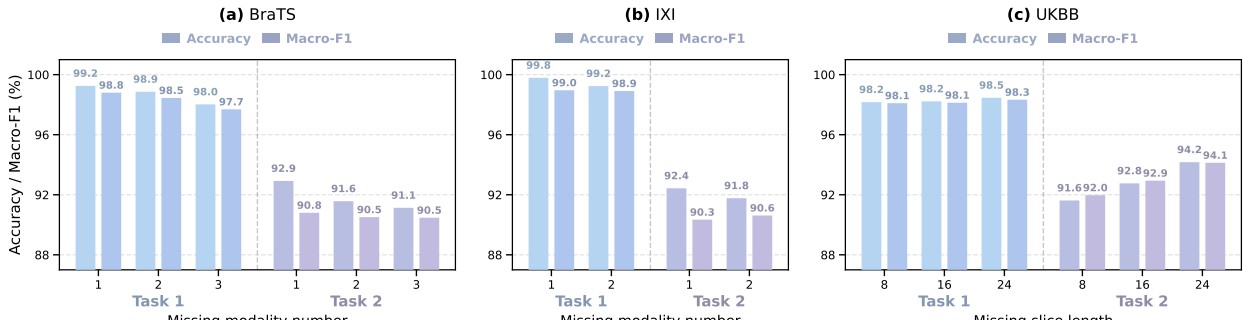

Figure 12: **Evaluation of completeness perception in CoPeVAE.** We report the accuracy and macro-F1 of Task 1 and Task 2 on BraTS, IXI, and UKBB. For BraTS and IXI, the numbers denote missing modality numbers; for UKBB, they denote missing slice lengths. Task 1 evaluates missing number/length detection, while Task 2 evaluates incompleteness positioning.

## C.7 Qualitative Analysis of Failure Cases

To further analyze the limitations of completeness perception, we visualize representative BraTS failure cases according to the correctness of Task 1 and Task 2 predictions. As shown in Fig. 14, the four cases reveal different effects of completeness perception errors.

- **Correct Count + Correct Position.** When both the missing count and position are correctly predicted, CoPeDiT produces anatomically plausible MRIs, preserving local structures and lesion details well.

- **Correct Count + Wrong Position.** When the count is correct but the position is wrong, the synthesized image remains structurally reasonable but shows modality-specific contrast drift, such as T1ce-like appearance when the target is T1.

- **Wrong Count + Correct Position.** When the count is wrong but the true missing modality is still covered by the predicted set, the model preserves the main target structure but introduces additional modality bias, leading to degraded local texture and lesion appearance.

- **Wrong Count + Wrong Position.** When both tasks are incorrect, the synthesized MRI suffers from the most severe degradation, including incorrect modality contrast, blurred anatomical boundaries, and larger reconstruction errors.

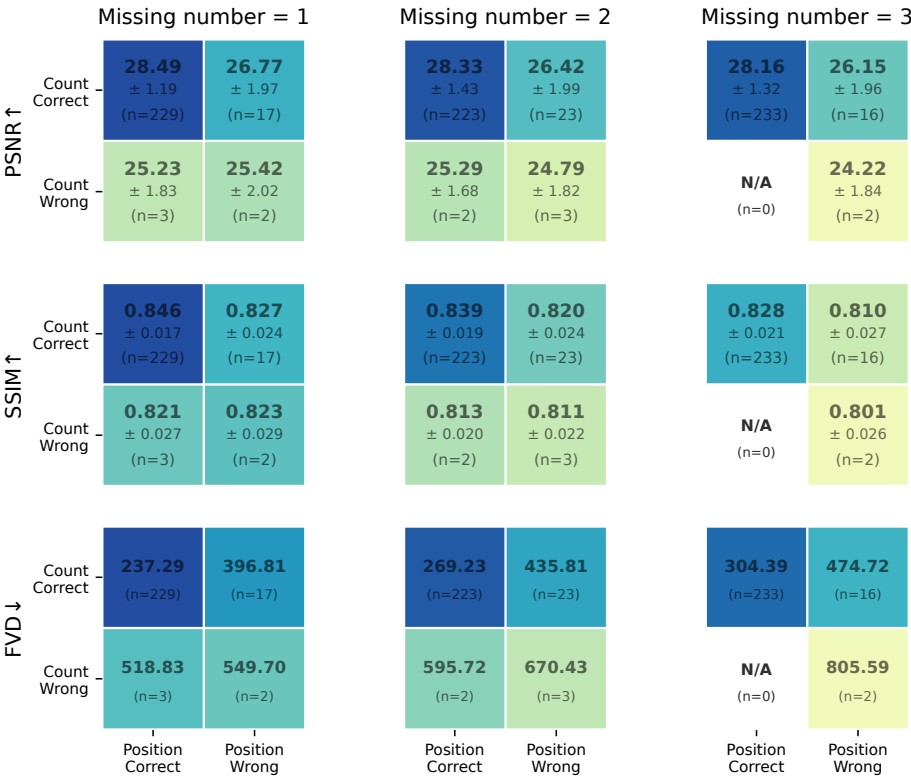

Figure 13: **Relationship between completeness-perception correctness and synthesis quality on the BraTS dataset**. We group test samples by the correctness of the CoPeVAE count and position predictions under different missing-modality settings, and report the corresponding synthesis metrics. For each heatmap, rows denote whether the predicted missing count is correct, columns denote whether the predicted missing position is correct. Cells with no samples are marked as N/A.

In summary, the failure patterns suggest that inaccurate modality semantics and miscalibrated missing-state severity are the main factors limiting synthesis quality when completeness perception is incorrect.

## C.8    Tumor Segmentation with Three Synthesized Modalities

In Sec. 4.4, we evaluate downstream tumor segmentation by replacing one missing modality with a synthesized one while keeping the other three modalities real. To further examine the downstream applicability of synthesized MRIs under a more challenging scenario, we additionally consider the case where only one real modality is available and the other three modalities are synthesized. Specifically, for each BraTS subject, we retain one modality from {T1, T1ce, T2, FLAIR} as the observed input and synthesize the remaining three modalities using each synthesis model. The completed four-modality inputs are then fed into the same multi-modal U-Net segmentation framework as used in Sec. 4.4. As reported in Table 21, replacing three modalities with synthesized ones is more challenging than the one-synthetic-modality setting, leading to lower segmentation accuracy for all methods. Nevertheless, CoPeDiT achieves satisfactory performance under these settings. The MRIs synthesized by CoPeDiT remain informative for downstream segmentation even when most of the multi-modal input is generated, further supporting the practical utility of our method under severe missing-modality scenarios.

We also provide qualitative examples corresponding to the downstream tumor segmentation experiment in Sec. 4.4. The qualitative results in Fig. 15 demonstrate that CoPeDiT yields segmentation masks more consistent with the ground-truth tumor regions under different missing-modality settings. In particular, the

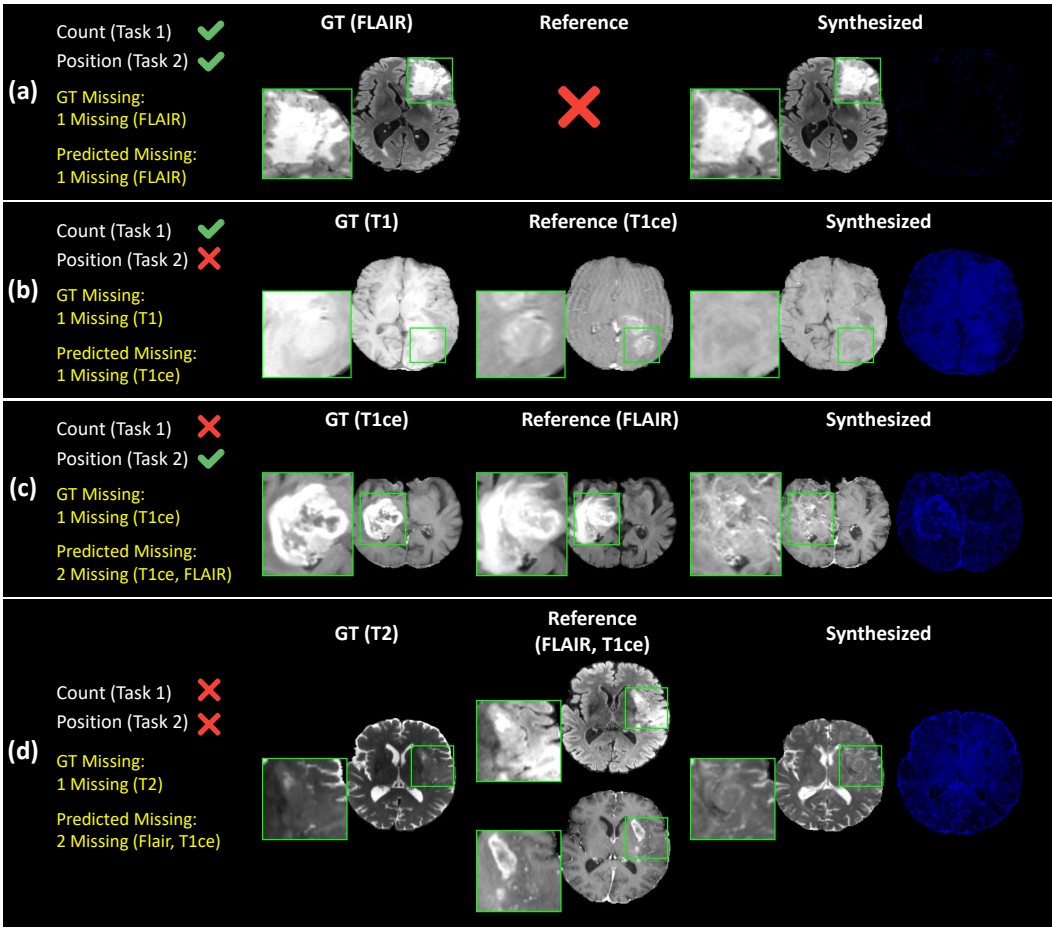

Figure 14: **Qualitative analysis of failure cases on the BraTS dataset.** We visualize representative cases grouped by the correctness of Count prediction (Task 1) and Position prediction (Task 2). (a) When both predictions are correct, CoPeDiT produces realistic synthesis with accurate local details. (b) Incorrect position prediction causes modality-specific semantic drift despite the correct missing count. (c) Incorrect count prediction introduces additional modality bias while still covering the true missing target. (d) When both predictions are incorrect, the synthesized MRI shows the largest degradation, including contrast mismatch, blurred boundaries, and stronger reconstruction errors.

predicted WT, TC, and ET regions better preserve tumor morphology and boundary continuity, leading to higher Dice scores in most representative cases.

## D    Discussion

**Cross-Modal Relationships Captured by Completeness Perception.** The cross-modal relationships learned by our CoPeDiT refer to data-driven anatomical and semantic correspondences among MRI modalities, rather than manually specified physical rules. For instance, different modalities share common brain anatomy and lesion locations, while exhibiting modality-specific contrast patterns, such as enhancement-related information in T1ce and hyperintense lesion regions in FLAIR. The proposed completeness-perception tasks encourage CoPeVAE to organize these relationships at multiple granularities: the count-focused prompt token captures the global degree of incompleteness, the position-focused prompt token identifies which modality or slice is missing, and the semantic prompt token models contextual compatibility between the observed and missing elements. In this sense, completeness perception is not simply generic latent conditioning, but a structured mechanism that distills cross-modal missing-state semantics into complementary prompt to-

Table 21: Downstream **tumor segmentation** results on the BraTS dataset when only one real modality is available and the remaining three modalities are synthesized. Results are averaged over four single-source settings, where T1, T1ce, T2, or FLAIR is retained as the only real input in turn.

| | Dice Score (%)↑ | | | |
|---|---|---|---|---|
| | WT | TC | ET | AVG |
| Missing | 83.75 | 82.81 | 83.50 | 83.35 |
| *GAN-based Methods* | | | | |
| MMGAN Sharma & Hamarneh (2020) | 86.90 | 86.27 | 85.26 | 86.14 |
| MMT Liu et al. (2023) | 87.41 | 85.92 | 84.58 | 85.97 |
| Hyper-GAE Yang et al. (2023) | 86.14 | 85.07 | 84.72 | 85.31 |
| *Diffusion Model-based Methods* | | | | |
| LDM Rombach et al. (2022) | 85.46 | 84.29 | 83.56 | 84.44 |
| ControlNet Zhang et al. (2023) | 85.87 | 84.70 | 83.80 | 84.79 |
| M2DN Meng et al. (2024) | 87.82 | 87.23 | 85.59 | 86.88 |
| DiffM$^4$RI Ye et al. (2026) | 88.17 | 86.76 | 85.83 | 86.92 |
| APT Shin et al. (2025) | 89.04 | **87.64** | 84.92 | 87.20 |
| ASLDM Zhou et al. (2026) | 88.49 | 86.67 | 84.72 | 86.63 |
| **CoPeDiT** | **89.26** | 87.49 | **86.35** | **87.70** |

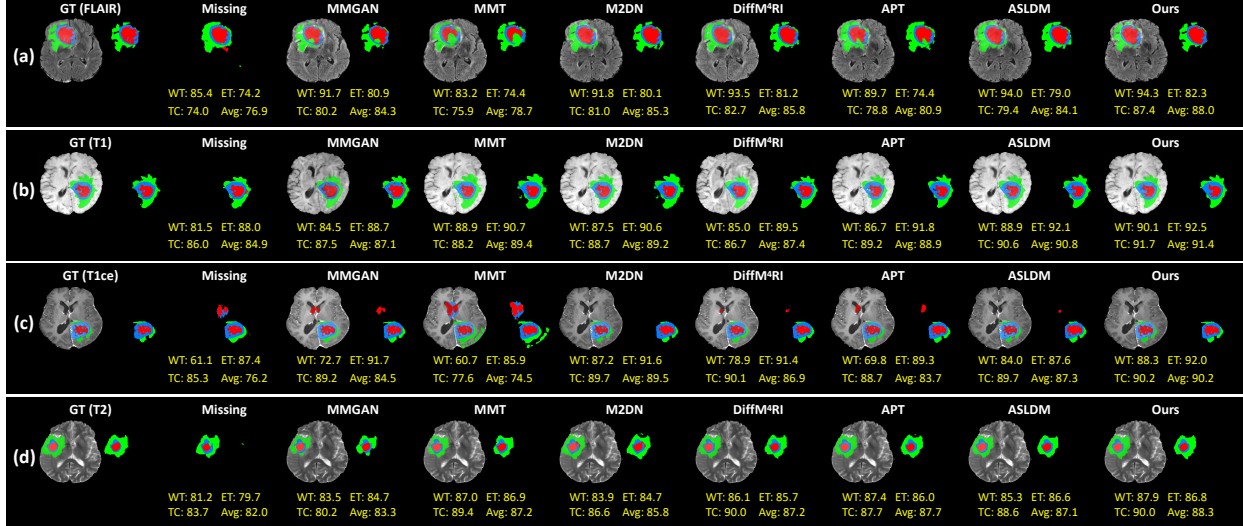

Figure 15: **Qualitative results of downstream tumor segmentation on the BraTS dataset.** Rows (a)–(d) exhibit representative cases where FLAIR, T1, T1ce, and T2 are missing and replaced by synthesized modalities, respectively. We visualize the corresponding segmentation predictions and report Dice scores for WT, TC, ET, and their average.

kens for diffusion guidance. The GradCAM, prompt-token distribution, and attention-map visualizations in Sec. 4.6 further indicate that the learned tokens attend to modality-discriminative regions, form separated missing-state representations, and guide modal-block attention toward the missing elements.

**Role and Scope of Completeness-Aware Prompt Tokens.** As reported in Fig. 1b and Table 4, the available latents already provide relatively strong anatomical information for conditional synthesis. Therefore, our completeness-aware prompt tokens are not intended to replace the available latents, but to complement them with compact and content-dependent missing-state guidance. To be specific, the count-focused prompt token summarizes the global severity of incompleteness, the position-focused prompt token encodes the missing modality/slice identity, and the semantic prompt token captures modality- or slice-specific contextual priors. Compared with binary mask codes, these learned prompt tokens are inferred from the

incomplete MRI itself and provide a more informative conditioning signal for modulating the task-relevant MDiT3D blocks. This interpretation is further supported by the visualization analysis in Sec. 4.6, where the learned prompt tokens exhibit semantically separable representations and guide attention toward task-relevant missing elements.

