# OpenReview forum: "Exploiting Completeness Perception with Diffusion Transformer for Unified 3D MRI Synthesis"
_TMLR — Under review for TMLR_

### Review · Reviewer_LTfC · 2026-03-18

**Summary Of Contributions:**

This paper addresses an important problem: missing modalities/data in medical imaging. The key gap found by the authors is that recent methods rely on external guidance or conditioning. The authors mentioned that these external inputs are not often available in real-world clinical settings. To address this limitation, this paper introduces a latent diffusion model with completeness perception for unified synthesis of 3D MRIs that automatically infers missing states. The proposed CoPeDiT framework consists of two main components: CoPeVAE, a tokeniser that learns completeness-aware prompt tokens through self-supervised pretext tasks designed to infer the number, position, and semantic characteristics of missing elements. MDiT3D, a diffusion transformer architecture tailored for volumetric MRI synthesis that integrates these learned prompts as conditional guidance during the diffusion process. The method is evaluated on multiple datasets and shows improvements over several methods.

The key strengths of this paper are that it addresses an important and practical problem in medical imaging, where missing modalities are frequent in clinical settings, and that the completeness-aware discriminative prompts serve as semantic guidance. The paper is well structured and organized. The authors conducted comprehensive evaluations, including ablation studies.

The main weakness is that the claim of a “unified” framework is somewhat overstated. The method relies on task-specific architectures, which limit the extent to which the framework is truly unified. The evaluation primarily focuses on image similarity metrics, while clinical or structural validation (e.g., volumetric or anatomical consistency analysis) is limited. The robustness of the method under unseen missing patterns, scanner variability, or noise conditions is not sufficiently analyzed.

**Additional Comments:**

Overall, this paper presents an interesting direction by introducing completeness-aware prompts for diffusion-based MRI synthesis. The integration of self-supervised pretext tasks with diffusion transformers is novel and technically sound. However, several claims could be better substantiated with stronger experimental validation and clearer positioning of the proposed framework relative to existing methods. With additional analysis on anatomical fidelity, robustness, and clearer methodological framing, the work could make a meaningful contribution to generative modelling for medical imaging.

**Audience:**

Yes

**Audience Explanation:**

The work combines several active research areas, including diffusion transformers, prompt learning, and self-supervised representation learning, which are of significant interest to the machine learning community. The idea of enabling generative models to self-perceive missing data states rather than relying on explicit conditioning is conceptually interesting and could inspire further research in generative modeling and incomplete data reconstruction.

Additionally, the application to 3D medical imaging synthesis broadens the impact of diffusion models beyond natural images and aligns with the growing interest in foundation models for biomedical data.

**Broader Impact Concerns:**

The work focuses on the synthetic generation of medical images, which may raise concerns about misuse or over-reliance on generated data in clinical decision-making. While the authors demonstrate improved synthesis quality, synthetic images should not replace real clinical data without proper validation.

It would be beneficial for the authors to discuss, potential risks of hallucinated anatomical structures, the importance of clinical oversight when using synthesized images and limitations in deploying such models in real healthcare workflows.

**Claims And Evidence:**

Yes

**Claims Explanation:**

The paper provides extensive quantitative comparisons across multiple datasets and reports improvements over several SOTA GAN-based and diffusion-based methods using standard metrics.  The inclusion of ablation studies for prompt tokens, pretext tasks, and transformer architecture design further supports the effectiveness of the proposed components.

However, some of the broader claims, particularly regarding robustness and structural consistency, are not fully supported by the presented experiments. The evaluation focuses primarily on pixel-level or perceptual metrics, which may not adequately capture anatomical correctness in medical imaging. Additional analyses, such as volume consistency, segmentation-based evaluation, or structural similarity metrics specific to anatomical regions, would strengthen the evidence supporting these claims.

Furthermore, the robustness of the method under realistic clinical variability (e.g., across scanners, domain shifts, or pathological variations) has not been thoroughly investigated.

**Requested Changes:**

1.	The paper currently describes CoPeDiT as a unified approach, but the model is trained separately on brain and cardiac MRI tasks. The authors should clarify whether the method is truly unified or rather a shared design paradigm with task-specific adaptations.
2.	Image similarity metrics alone are insufficient for evaluating medical image synthesis. The authors should consider additional evaluations, such as including metrics NMSE and MAE, region-wise or lesion-wise structural similarity or volumetric comparisons, Segmentation-based validation across multiple structures and anatomical consistency metrics.
3.	Qualitative figures should show where to look into in zoomed areas, and it would be better to add error maps for all qualitative figures. Include all modalities in the qualitative figures, not just one specific modality.
4.	The paper would benefit from experiments assessing robustness to, unseen missing modality combinations, domain shifts (e.g., different datasets or scanners or multi-site), noise or corrupted inputs.
5.	Compare against additional recent diffusion-based medical synthesis methods, particularly those designed for volumetric or multimodal data.
6.	Provide more qualitative analysis of failure cases. Understanding when the model fails to synthesise realistic MRIs would provide insights into the limitations of the completeness perception mechanism.
7.	Discuss computational efficiency more clearly, especially considering the complexity of diffusion transformers in 3D medical imaging.
8.	Although the effectiveness of the pretext tasks in improving overall synthesis performance has been validated, it is important to also evaluate the performance of the prompt encoders. Their classification accuracy directly influences the synthesis process, particularly in correctly identifying missing contrasts and their semantic relationships. Therefore, a more thorough validation and discussion of this component is necessary.
9.	As shown by the evaluation and results, the pre-text tasks provide a proper semantic prior to guide the synthesis process, providing global and finer-level high-frequency information that directly captures these discrete cues across modalities. So, rather than framing the contribution primarily around the problem with a defined mask or guidance, it would be more accurate to emphasise this semantic guidance and the self-supervision provided by the pre-text tasks, which eventually overcome the potential barriers posed by mask codes.
10.	Also, it’s unclear how the limitations mentioned in the introduction section directly affect the use of the defined guidance for identifying missing contrasts. Since the model is trained on a fixed number of modalities/contrasts, how the implicit indicators as guidance overcome such limitations should be clearly discussed.
11.	In relation to the above pre-text-related clarifications, it’s better to include an ablation study on replacing pp with a mask, to see how surrounding pre-text tasks with a known prior on missing contracts would actually benefit, and provide clear motivation for the main claims.
12.	Add qualitative experimental results on downstream segmentation analysis. Rather than using one synthetic modality, the downstream should be evaluated by replacing all three with synthetic ones, for better understanding of their downstream applicability.
13.	Missing code reproducibility section. Consider adding an anonymous repository link.

---

> ### Author Response · Authors · 2026-06-19
> **Respond Part 1**
>
> We would like to thank Reviewer LTfC for the insightful comments.
>
> **Q1 (“Unified” Framing):** Thank you for this helpful feedback. We have clarified that CoPeDiT is not a single jointly trained model across brain and cardiac MRI domains. Instead, our “unified” refers to a shared missing-data synthesis framework with a common prompt-based conditioning and diffusion generation pipeline, while brain missing-modality synthesis and cardiac missing-slice synthesis are trained as separate task-specific instantiations. This clarification has been added to the Introduction (page 2) and Implementation Details (page 8), and the Abstract, contribution statement (page 3), notation section (Sec. 3.1 on page 4), and Conclusion (page 12) have also been revised to consistently reflect this framing.
>
> **Q2 (Additional Metrics):**
> * We agree that global similarity metrics alone are insufficient, and have therefore added MAE to all main quantitative tables and further included segmentation-based anatomical consistency metrics in Appendix C.1 (page 21 and 22). Specifically, using a frozen nnU-Net trained on real images, CoPeDiT achieves higher Dice and lower HD95 than baselines.
>
> * Regarding volumetric evaluation, our original evaluation already includes FVD, where each 3D MRI volume is treated as a video and axial slices are treated as frames, thereby assessing volume-level spatial/structural consistency across slices.
>
> | Setting              |        Method | Lesion MAE↓ | Lesion SSIM↑ | Dice↑ | HD95↓ |
> | -------------------- | ------------: | ----------: | -----------: | ----: | ----: |
> | 1 missing modality   | Best baseline |       0.073 |        0.810 | 87.08 |  8.29 |
> | 1 missing modality   |       CoPeDiT |       0.067 |        0.815 | 87.63 |  7.98 |
> | 3 missing modalities | Best baseline |       0.086 |        0.799 | 83.48 | 10.57 |
> | 3 missing modalities |       CoPeDiT |       0.078 |        0.802 | 85.09 |  9.19 |
>
>
>
> **Q3 (Qualitative Figures):** Following your suggestion, we have revised Figs. 4 (page 7) and 5 (page 8) to include all target modalities with zoomed regions and corresponding error maps, and also added error maps to the cardiac MRI visualization in Fig. 9 (page 22). In addition, we provide new qualitative results in Figs. 10 (page 23) and 11 (page 24) on BraTS and IXI, respectively, covering all missing-modality combinations.
>
> **Q4 (Robustness Evaluation):** Thank you for the suggestion. We have added a new robustness experiment in Appendix C.5 (page 24), where selected missing modality combinations are excluded during training and used only for testing. CoPeDiT consistently performs well under these unseen incomplete configurations, supporting the robustness of our completeness-aware prompts beyond observed missing patterns.
>
> | # Missing | Available Modalities | PSNR↑ | SSIM↑ |  MAE↓ |  FID↓ |   FVD↓ |
> | --------: | -------------------- | ----: | ----: | ----: | ----: | -----: |
> |         1 | T1, T1ce, T2         | 28.10 | 0.839 | 0.057 | 14.19 | 302.16 |
> |         2 | T1, FLAIR            | 27.92 | 0.827 | 0.059 | 15.98 | 357.83 |
> |         3 | T2                   | 27.20 | 0.817 | 0.065 | 18.75 | 458.63 |
>
>
> **Q5 (Additional Baselines):** As suggested, we have added two recent diffusion-based medical synthesis baselines, APT (CVPR 2025) [1] and ASLDM (TMI 2025) [2], to all main quantitative comparisons on BraTS, IXI, and UKBB. CoPeDiT consistently achieves better performance across missing configurations, providing a stronger comparison with recent volumetric/multimodal synthesis methods.

---

> ### Author Response · Authors · 2026-06-19
> **Respond Part 2**
>
> **Q6 (Analysis of Failure Cases):** We thank the reviewer for this valuable feedback. We have added Appendix C.7 (page 26), where we visualize representative BraTS cases under four Task 1/Task 2 correctness settings. This analysis shows that incorrect position prediction mainly causes modality-specific contrast drift and local texture degradation, while incorrect count prediction miscalibrates the missing-state severity. The most severe artifacts occur when both predictions are incorrect, clarifying the limitations of the completeness perception mechanism.
>
> **Q7 (Computational Efficiency):** We thank the reviewer for this helpful suggestion. Following your suggestion, we have revised Section 4.5 (page 10) to discuss computational efficiency more explicitly, reporting parameters, FLOPs, wall-clock training time, inference latency, and VRAM usage under standardized settings. The revised analysis shows that the completeness-perception tasks introduce only marginal training overhead, while CoPeDiT has comparable inference cost to vanilla DiT-3D and remains faster than heavier diffusion baselines, providing a clearer view of the accuracy-efficiency trade-off.
>
> **Q8 (Pretext-task Accuracy):** Thank you for pointing this out. We agree that the prompt encoders should be validated more explicitly. In the revised manuscript, Appendix C.6 (page 25) now evaluates the CoPeVAE prompt encoders using Task 1/2 accuracy and macro-F1 on held-out data, demonstrating reliable recognition of both missing-state severity and missing positions. We also include a correctness-conditioned synthesis analysis, which shows that synthesis quality is closely linked to the correctness of completeness perception, further supporting the effectiveness of the learned prompts as guidance for MDiT3D.
>
> **Q9 (Contribution Framing):** We thank the reviewer for this suggestion and giving us the opportunity to modify our contribution statement. We have revised the contribution statements (page 3) to better emphasize the core role of self-supervised pretext tasks in learning semantic prompts, including global missing severity, fine-grained missing positions, and modality/slice-specific semantic priors, rather than mainly framing the contribution around the limitations of mask codes.
>
> **Q10 (Scope and Inference Protocol of Implicit Guidance):** Thank you for highlighting this issue. We have clarified the scope and inference protocol in Sec. 3.2 on page 5. The binary missing labels are used only as supervision for the self-supervised pretext tasks during CoPeVAE training, where missing patterns are synthetically generated from complete samples. At inference, no binary mask codes are manually provided; instead, the frozen CoPeVAE automatically extracts content-dependent prompt tokens from the observed incomplete MRI. We also clarify that this design operates within the predefined modality/slice vocabulary, and is intended to reduce reliance on hand-crafted inference-time mask codes rather than to handle entirely unseen contrast types outside the training protocol.
>
> **Q11 (Additional Ablation):** Thank you for your constructive comment. We have added an additional ablation by replacing the learned positioning prompt $\mathbf{p}^p$ with explicit mask codes while keeping the other prompts unchanged, as reported in Table 4 (page 8) and Table 16 (page 23). As reported, mask codes improve over removing $\mathbf{p}^p$, but still underperform the learned prompt, supporting that the pretext-driven $\mathbf{p}^p$ provides more informative guidance than a hand-crafted missing-position prior.
>
> | Dataset |                               Variant | Avg. PSNR↑ | Avg. MAE↓ | Avg. FVD↓ |
> | ------- | ------------------------------------: | ---------: | --------: | --------: |
> | BraTS   |                    w/o $\mathbf{p}^p$ |      26.47 |     0.068 |    468.28 |
> | BraTS   | $\mathbf{p}^p \rightarrow$ Mask Codes |      27.20 |     0.062 |    439.40 |
> | BraTS   |                               CoPeDiT |      28.10 |     0.059 |    288.49 |
> | IXI     |                    w/o $\mathbf{p}^p$ |      22.40 |     0.100 |   1274.51 |
> | IXI     | $\mathbf{p}^p \rightarrow$ Mask Codes |      23.74 |     0.083 |    762.46 |
> | IXI     |                               CoPeDiT |      24.13 |     0.072 |    643.88 |
> | UKBB    |                    w/o $\mathbf{p}^p$ |      24.78 |     0.082 |    616.16 |
> | UKBB    | $\mathbf{p}^p \rightarrow$ Mask Codes |      25.36 |     0.080 |    515.31 |
> | UKBB    |                               CoPeDiT |      25.91 |     0.075 |    404.60 |

---

> ### Author Response · Authors · 2026-06-19
> **Respond Part 3**
>
> **Q12 (Segmentation Analysis):**
> * Following your suggestion, we added qualitative downstream tumor segmentation results in Appendix C.8 (page 29), corresponding to the experiment in Sec. 4.4, to visually compare segmentation masks under different missing-modality settings.
>
> * As suggested, in Appendix C.8 (page 27), we also added a more challenging downstream evaluation where only one real modality is retained and the remaining three modalities are synthesized, directly addressing the suggested three-synthetic-modality setting. These additional results further demonstrate the downstream applicability of the synthesized MRIs under both standard and severe missing-modality scenarios.
>
> **Q13 (Code Reproducibility):** Thank you for the feedback. We have added an anonymous code repository link in the abstract: https://anonymous.4open.science/r/CoPeDiT-B0E9/.
>
> **Q14 (Broader Impact Concern):** Thank you for raising this important concern. We agree that synthetic medical images should not be treated as replacements for real clinical scans, since generative models may introduce hallucinated or imperfect anatomical details. We have therefore added a statement in the Limitations section on page 12 to clarify that synthetic MRIs should be used only as auxiliary data and require expert validation before any clinical use.
>
> **References:**
>
> [1] Shin Y, Lee Y, Jang H, et al. Anatomical consistency and adaptive prior-informed transformation for multi-contrast mr image synthesis via diffusion model. Proceedings of the Computer Vision and Pattern Recognition Conference. 2025: 30918-30927.
>
> [2] Zhou L, Qu X, Fu T, et al. Anatomy-aware Sketch-guided Latent Diffusion Model for Orbital Tumor Multi-Parametric MRI Missing Modalities Synthesis. IEEE Transactions on Medical Imaging, 2025.

---

### Review · Reviewer_EqfE · 2026-03-30

**Summary Of Contributions:**

This paper addresses 3D MRI synthesis under incomplete observations, including missing modalities in brain MRI and missing slices in cardiac MRI. It proposes a unified two-stage framework, CoPeDiT, that replaces explicit mask-based missing-state indicators with learned completeness-aware conditioning. In the first stage, a VQGAN-style 3D autoencoder, CoPeVAE, is trained with auxiliary pretext tasks to encode missing-count, missing-identity, and semantic information into learned conditioning embeddings. In the second stage, these embeddings are injected into a task-adapted diffusion transformer, MDiT3D, for conditional synthesis. The paper reports strong quantitative results across three datasets, along with ablations, visualization analyses, and a downstream tumor segmentation experiment.

The main strengths are the unified treatment of two related MRI completion settings, strong empirical results, and a reasonably well-designed set of ablations. The main weaknesses are that the claimed “completeness perception” is implemented mainly through learned latent conditioning rather than a fundamentally new missingness modeling mechanism, and the paper does not yet fully clarify what cross-modal relationships are actually learned or why the proposed conditioning is superior beyond empirical performance.

**Audience:**

Yes

**Audience Explanation:**

The paper addresses a practical and important problem in medical image synthesis, namely missing-modality and missing-slice reconstruction in 3D MRI. Its combination of learned conditioning, diffusion-based synthesis, and evaluation on multiple datasets would likely be of interest to at least part of the TMLR audience, especially researchers working on medical image generation, conditional generative modeling, and representation learning for incomplete data.

**Claims And Evidence:**

Yes

**Claims Explanation:**

The paper provides generally strong empirical evidence for its main performance claims, including consistent improvements over multiple baselines, comprehensive ablations, and a downstream tumor segmentation study. However, some of the higher-level conceptual claims are not fully supported with equally clear evidence. In particular, the proposed “completeness perception” is implemented mainly through learned latent conditioning with auxiliary objectives, so the mechanism by which cross-modal missingness is modeled remains somewhat unclear. Similarly, claims about capturing long-range, anisotropic, or irregular dependencies appear stronger than what is directly validated by the analysis. Overall, the evidence is convincing for empirical effectiveness, but only partially convincing for some of the broader conceptual claims.

**Requested Changes:**

1. The current evaluation is dominated by global similarity and perceptual metrics (PSNR, SSIM, FID, and FVD). PSNR and SSIM mainly reflect global average similarity, while FID and FVD are more distribution-oriented and perception-oriented. As a result, a model may still obtain favorable overall scores even when most normal tissues are synthesized well but small lesions, lesion boundaries, or enhancing regions are blurred. This is particularly concerning in medical imaging, where clinical value often depends precisely on the faithful preservation of such local abnormalities. The downstream tumor segmentation experiment is a useful step toward clinical validation, but it remains an indirect proxy. The paper would be stronger if it also included direct lesion-region fidelity metrics, since global reconstruction quality and downstream utility do not fully characterize pathological accuracy.
2. The method assumes that common MRI sequences (e.g., T1, T2, and FLAIR) share learnable cross-modal relationships that allow missing modalities to be inferred from observed ones. However, the proposed “completeness perception” is implemented mainly as a VQGAN-style latent encoder with auxiliary classification and contrastive objectives that produce learned conditioning embeddings for diffusion. While effective in practice, the paper does not sufficiently explain what cross-modal relationships are actually captured, making the method appear closer to black-box learned latent conditioning than to a genuinely new mechanism for modeling missingness.
3. The relatively strong performance of the “w/o Mask Code” setting in Figure 1b suggests that the available latents themselves already provide strong conditional information. Therefore, the paper should clarify more explicitly what additional information the learned prompts contribute beyond these existing conditions, rather than simply serving as a replacement for mask codes.
4. Writing and terminology:

1). On page 2, the paper states: “We hypothesize that, for diffusion models, such self-guided prompts may serve as an effective alternative to manually defined masks, and potentially offer even stronger guidance signals”. Since Figure 1b is already presented as empirical evidence supporting this claim, the wording “hypothesize” seems unnecessarily tentative. A phrasing such as “our results suggest” or “our results show” may be more appropriate.

2). The paper states that MDiT3D captures the “complex, long-range dependencies” in high-dimensional 3D MRIs. However, it is unclear where such “long-range” dependencies are explicitly modeled. More importantly, the paper does not appear to provide dedicated mechanisms specifically addressing long-range, anisotropic, or irregular dependencies. As written, these claims in the introduction may be somewhat overstated.

3). The paper uses “learned prompts” “prompts” and “prompt tokens” to refer to the conditioning inputs of the DiT backbone. This terminology may be somewhat imprecise and potentially confusing, since “prompt” is more commonly associated with textual input. Terms such as “conditioning embeddings” “conditioning tokens” or “condition-guided generation” may be more accurate and less ambiguous.

4). On page 3, in the sentence “We propose a unified formulation with task-specific instantiations, dubbed CoPeDiT, for both 3D brain and cardiac missing MRI synthesis under arbitrary incomplete scenarios, without the need for explicit external indicators as guidance” the phrase “unified formulation” may not be the most appropriate. The paper appears to propose a unified framework or method with task-specific variants, rather than merely a formulation.

---

> ### Author Response · Authors · 2026-06-19
>
> Thank you for your valuable suggestions.
>
> **Q1 (Lesion-region Fidelity Metrics):** Thank you for the valuable suggestion. We have added a new Appendix C.1 (page 21 and 22), where we use BraTS ground-truth tumor annotations to compute lesion-wise MAE and SSIM within WT, TC, and ET regions. The additional evaluation show that CoPeDiT achieves better lesion-region fidelity than the strongest baselines, e.g., reducing lesion-wise MAE from 0.059/0.074 to 0.057/0.068 under 1/3 missing modalities.
>
> **Q2 (Cross-modal Relationships):** We thank the reviewer for raising this important point.
>
> * We agree that the cross-modal relationships captured by completeness perception should be explained more explicitly. In our design, these relationships refer to data-driven anatomical and semantic correspondences across MRI modalities, such as shared brain structures, lesion locations, and modality-specific contrast patterns that are informative for inferring the missing modality. The visualization analysis in Sec. 4.6 (page 11) provides supporting evidence, where the salient regions indicate that different pretext tasks attend to global anatomy and modality-discriminative local patterns, the prompt-token distributions show decoupled missing-count and missing-identity semantics, and the attention maps demonstrate that prompt tokens guide modal blocks toward the actual missing elements.
>
> * We have added a discussion in Appendix D on page 28 to clarify that completeness perception is not merely black-box latent conditioning, but a structured way to distill cross-modal missing-state semantics into count-, position-, and semantic-level prompt tokens for diffusion guidance.
>
> **Q3 (Additional Contribution of Prompt Tokens):** We thank the reviewer for this insightful suggestion.
>
> * We agree that the available latents already provide strong anatomical conditioning. Our learned prompt tokens are designed to complement these latents by providing distilled missing-state guidance, including missing severity (Task 1), missing identity/position (Task 2), and modality/slice-specific semantic priors (Task 3). As clarified in the Introduction (page 2), this advantage comes from encouraging the tokenizer to perceive global and local anatomical structures and lesion patterns at coarse and fine levels, thereby enabling more semantically coherent generation. This mechanism is further supported by the visualization analysis in Sec. 4.6 (page 11), where the learned prompt tokens show semantically separable representations and guide attention toward the actual missing elements.
>
> * We have also added a discussion in Appendix D on page 29 to further clarify the role and scope of completeness-aware prompt tokens.
>
> **Q4 (Writing and Terminology):**
>
> **1) “Hypothesize” Phrasing:** Thank you for pointing this out. We have revised the wording on page 2 from “We hypothesize” to “Our empirical results suggest”, making the statement more consistent with the empirical evidence presented in Fig. 1b.
>
> **2) “Long-range Dependencies” Statement:** We agree that the previous wording could overstate the architectural claim. We have softened and clarified the description (page 2 and page 3) by replacing the broad “long-range, anisotropic, and irregular dependencies” statement with more specific task-relevant dependencies, including 3D spatial context, inter-modal relationships in brain MRI, and through-plane continuity in cardiac MRI.
>
> **3) “Prompts” Terminology:** We appreciate the reviewer’s careful reading of our manuscript. To avoid ambiguity, we have standardized the terminology throughout the manuscript as “prompt tokens” or “completeness-aware prompt tokens”, and clarified that these tokens are learned by CoPeVAE through self-supervised pretext tasks and then used as conditioning inputs for MDiT3D. We keep the term “prompt tokens” rather than replacing it entirely with “conditioning tokens”, because they encode explicit completeness semantics beyond generic conditioning embeddings.
>
> **4) “Unified” Framing:** We agree that the phrase “unified formulation” could be misleading, as CoPeDiT is better described as a unified framework with task-specific variants. We have revised the wording accordingly throughout our manuscript. Specifically, in the Abstract, we now describe CoPeDiT as “a shared completeness-perception framework for 3D MRI synthesis” with “task-specific instantiations”. In the Introduction (page 2) and the first contribution (page 3), we replaced “unified formulation” with “unified framework”. We also revised Sec. 3.1 (page 4) to present the two tasks under a common notation rather than claiming a unified formulation.

---

### Review · Reviewer_Hi66 · 2026-06-06

**Summary Of Contributions:**

CoPeDiT is a two-stage latent diffusion framework for 3D MRI synthesis under missing data (missing brain modalities or missing cardiac slices). The central idea is that the model should infer the missing state itself rather than rely on externally supplied binary mask codes: a VQGAN-based tokenizer, CoPeVAE, is trained with three self-supervised pretext tasks (missing-count detection, position identification, and an inter-modal/slice contrastive task) to produce three "completeness-aware prompt" tokens, and a 3D diffusion transformer, MDiT3D, with task-specific alternating blocks then injects these prompts via adaLN to guide synthesis, adding noise only to the missing regions. The method is evaluated on BraTS, IXI, and a combined cardiac set against seven baselines, with ablations, a downstream tumor-segmentation experiment, efficiency analysis, and prompt sensitivity/intervention studies.

**Audience:**

Yes

**Audience Explanation:**

Missing data in the medical field is an important problem.

**Claims And Evidence:**

Yes

**Claims Explanation:**

## Strengths

1. **The core question is well-motivated.** The shift from explicit mask codes to inferred completeness prompts is cleanly framed, and the argument that mask codes only mark location without characterizing the actual incomplete state is reasonable and clearly stated.

2. **The experimental range is broad.** Beyond the main comparisons, the paper includes prompt-component ablations, an injection-position study, a backbone study, a downstream segmentation task, efficiency/latency reporting, and prompt sensitivity and intervention analyses. The plug-and-play experiment (prompts improving other architectures over their native mask codes) is especially valuable, since it supports the central claim somewhat independently of the specific MDiT3D design.

## Weaknesses

1. **The pretext-task accuracy is never reported, and the whole method depends on it.** Appendix B.2 clarifies that mask codes are used to supervise the pretext tasks during pretraining, and at inference the entire guidance signal flows through CoPeVAE correctly inferring the missing state. Yet the paper never reports how accurately the count and position classifiers actually perform on held-out data. Since everything downstream rests on these inferences, their standalone accuracy is the most important missing number.

2. **Results lack statistical support and rely on a single split.** Several margins over the strongest baselines fall within one standard deviation, FID and FVD have no error bars at all, and all results come from one train/test split with no multiple seeds or cross-validation. As written, "significantly outperforms" is colloquial rather than statistical, and method gains can't be cleanly separated from split variance.

**Requested Changes:**

Please refer to the Weakness.

---

> ### Author Response · Authors · 2026-06-19
>
> Thank you for your constructive feedback.
>
> **Q1 (Pretext-task Accuracy):** We appreciate the important suggestion and we have added a new Appendix C.6 (page 25) to report the held-out performance of CoPeVAE’s pretext-task classifiers. Specifically, Fig. 12 reports accuracy and macro-F1 for Task 1 and Task 2 across BraTS, IXI, and UKBB, showing around 98%-99% performance for missing number/length detection and over 90% performance for missing position localization. Additionally, we further add Fig. 13 to relate classifier correctness to synthesis quality on BraTS, showing that correct count and position predictions lead to the best synthetic performance (e.g., PSNR/SSIM/FVD), while prediction errors cause clear degradation.
>
> **Q2 (Statistical Support):**
> We thank the reviewer for this insightful comment. We have addressed it from three aspects.
>
> * We added a new Appendix C.2 (page 22), where we evaluate CoPeDiT and representative diffusion baselines under three random seeds on two challenging settings: BraTS with three missing modalities and IXI with two missing modalities. The results show that our method maintains consistent advantages across seeds, suggesting that the gains are not caused by a single random run, although we agree that this focused analysis does not replace exhaustive cross-validation.
>
> * Regarding FID and FVD, they are distribution-level generative metrics computed from sets of generated and real samples, rather than per-subject metrics like PSNR and SSIM. Therefore, subject-level standard deviations are not directly defined and are commonly omitted in generative-model evaluation. Nevertheless, our added multi-seed analysis now reports run-to-run standard deviations for FID and FVD.
>
> * We also revised the manuscript wording to avoid unsupported statistical claims. In particular, we replaced phrases such as "significantly outperforms" with more precise descriptions such as "consistently improves upon" or "achieves better average performance" and softened related statements throughout the abstract and contribution (page 3).

---

### Author Response · Authors · 2026-06-19
**General Statement**

We sincerely thank all reviewers for their constructive and valuable comments. We are encouraged that the reviewers recognized the importance of the missing-data problem in 3D MRI synthesis, the motivation of moving beyond manually specified mask codes, and the comprehensive experimental validation of our framework.
In response to the reviewers’ comments, we have substantially revised and strengthened the manuscript. The major updates include:

* **Clarifying the scope of CoPeDiT.** We revised the wording throughout the manuscript to clarify that CoPeDiT is a shared completeness-perception framework with task-specific instantiations for brain missing-modality synthesis and cardiac missing-slice synthesis, rather than a single jointly trained model across domains.

* **Strengthening medical and structural validation.** We added MAE to the main quantitative tables and included additional lesion-wise fidelity and anatomical consistency evaluations, including lesion-region MAE/SSIM and segmentation-based Dice/HD95 metrics.

* **Providing stronger evidence for completeness perception.** We added held-out evaluations of the CoPeVAE pretext-task classifiers and further analyzed how missing-state prediction correctness affects synthesis quality.

* **Improving robustness and statistical support.** We added multi-seed robustness analysis and an unseen missing-modality combination experiment to better support the stability and generalization of the proposed prompt tokens.

* **Expanding qualitative and failure-case analyses.** We revised the qualitative figures with all target modalities, zoomed regions, and error maps, and added a dedicated failure-case analysis to clarify when completeness perception errors affect synthesis.

* **Enhancing comparisons, ablations, and efficiency analysis.** We added recent diffusion-based baselines, included a prompt-token versus mask-code ablation, evaluated downstream segmentation with three synthesized modalities, and revised the computational efficiency discussion with detailed cost comparisons.

We have also revised terminology, softened overstated claims, added an anonymous code link, and updated the manuscript text accordingly, with all changes highlighted in blue. We believe these revisions directly address the reviewers’ concerns and improve the clarity, rigor, and reliability of the manuscript.